# SARS-CoV-2-specific cellular and humoral immunity after bivalent BA.4/5 COVID-19-vaccination in previously infected and non-infected individuals

Rebecca Urschel[1], Saskia Bronder [1], Verena Klemis[1], Stefanie Marx[1], Franziska Hielscher[1], Amina Abu-Omar[1], Candida Guckelmus[1], Sophie Schneitler [2], Christina Baum[3], Sören L. Becker [2], Barbara C. Gärtner [2], Urban Sester[4], Leonardo Martinez [5], Marek Widera [6], Tina Schmidt [1] & Martina Sester [1,7] ✉

Knowledge is limited as to how prior SARS-CoV-2 infection influences cellular and humoral immunity after booster-vaccination with bivalent BA.4/5-adapted mRNA-vaccines, and whether vaccine-induced immunity may indicate subsequent infection. In this observational study, individuals with prior infection ($n = 64$) showed higher vaccine-induced anti-spike IgG-antibodies and neutralizing titers, but the relative increase was significantly higher in non-infected individuals ($n = 63$). In general, both groups showed higher neutralizing activity towards the parental strain than towards Omicron-subvariants BA.1, BA.2 and BA.5. In contrast, CD4 or CD8 T cell levels towards spike from the parental strain and the Omicron-subvariants, and cytokine expression profiles were similar irrespective of prior infection. Breakthrough infections occurred more frequently among previously non-infected individuals, who had significantly lower vaccine-induced spike-specific neutralizing activity and CD4 T cell levels. In summary, we show that immunogenicity after BA.4/5-bivalent vaccination differs between individuals with and without prior infection. Moreover, our results may help to improve prediction of breakthrough infections.

SARS-CoV-2 variants of concern (VOCs) such as Omicron (B.1.1.529) have shown increased escape from neutralizing antibodies, which reduces the ability to prevent infection[1–3]. Together with waning immunity, this led to a substantial increase in the incidence of SARS-CoV-2 infections[4]. As a result, bivalent mRNA booster vaccines encoding the spike proteins of the ancestral WA1/2020 strain and of either the Omicron BA.1 or BA.5 sublineages have been developed[5,6]. Most available immunogenicity studies have largely been restricted to neutralizing antibody activity[6–10]. Neutralizing antibody titers were shown to increase after vaccination with the bivalent vaccine to a

[1]Department of Transplant and Infection Immunology, Saarland University, 66421 Homburg, Germany. [2]Institute of Medical Microbiology and Hygiene, Saarland University, 66421 Homburg, Germany. [3]Occupational Health Care Center, Saarland University, 66421 Homburg, Germany. [4]Department of Nephrology, SHG-Klinikum Völklingen, 66333 Völklingen, Germany. [5]Boston University, School of Public Health, Department of Epidemiology, Boston, MA, USA. [6]Institute for Medical Virology, University Hospital Frankfurt, Goethe University Frankfurt, Frankfurt, Germany. [7]Center for Gender-specific Biology and Medicine (CGBM), Saarland University, 66421 Homburg, Germany. ✉e-mail: martina.sester@uks.eu

slightly larger or similar extent as after monovalent vaccination, and the titers against the ancestral strain remained higher than against the Omicron strains[7–9].

Despite the increase in infections in the Omicron era, the incidence of severe disease remained considerably low in otherwise healthy individuals[1,2,11]. This may be due lower virulence of the Omicron subvariants, to an increasing number of individuals with hybrid immunity and/or to the fact that SARS-CoV-2 specific T cells, which are discussed to have a potential role in protection from severe disease[12,13], are less affected by mutations in the VOCs spike protein. Up to now, knowledge on the induction of spike-specific CD4 and CD8 T cells and on the impact of previous infection on immunogenicity after bivalent vaccination is limited, as most studies have reported aggregated data with small sample sizes[8,9]. This knowledge is becoming increasingly relevant, as more and more individuals undergo infection with SARS-CoV-2. The aim of this observational study was therefore to characterize the differences between individuals with and without previous infection in terms of spike-specific IgG, neutralizing activity and CD4 and CD8 T cells against the ancestral spike and Omicron subvariants before and after BA.4/5 bivalent vaccination. So far, most studies have focused on the role of humoral immunity for protection, whereas much less attention has been given to vaccine-induced cellular immunity[14]. Therefore, both study participants with and without prior infection were followed up to collect information on the association of individual vaccine-induced antibody or T cell levels with subsequent breakthrough infection.

Here we show that immunogenicity of the bivalent vaccine differs between individuals with and without prior infection. Moreover, breakthrough infections were more frequent among previously uninfected individuals, who had significantly lower vaccine-induced neutralizing antibodies and CD4 T cell levels.

## Results

### Study population

We recruited 127 immunocompetent individuals (49.5 ± 13.5 years, 42 males, 85 females) who underwent COVID-19 vaccination with a bivalent COVID-19 vaccine (Comirnaty Original/Omicron BA.4/5, BioNTech/Pfizer), as per German regulations. Among them, 64 were grouped as previously infected either by self-reported history of SARS-CoV-2 infection (confirmed by PCR or rapid antigen test) and/or by a positive nucleocapsid protein (NCP) serology. Sixty-three individuals were grouped as non-infected based on self-reporting and negative NCP-serology (Table 1). Individuals either had 3 or 4 immunization events (mostly 3 vaccinations with and without 1 infection) before receiving the bivalent vaccine. An overview of the study design is shown in figure S1. Most individuals had received prior vaccinations with an mRNA vaccine, and a minor part had a history of heterologous vector/mRNA vaccinations. Individuals with prior infection were younger, and most infections had occurred in the Omicron BA.2 era. Blood samples were drawn before and 14 (IQR 3) days after vaccination to determine differential blood counts and vaccine-induced humoral and cellular immunogenicity. Demographic characteristics and differential blood counts are shown in Table 1. General leukocyte and lymphocyte numbers did not differ between previously infected and non-infected individuals, except for monocytes which were significantly lower among individuals with prior infection (Table 1).

### Adverse events after bivalent vaccination

Adverse events were analyzed in the first week after the bivalent vaccination based on self-reporting using a questionnaire. Approximately 80% of individuals reported local or systemic adverse events or both (Fig. 1a) with no differences between individuals with and without prior infection ($p = 0.872$). Adverse events were overall mild, and pain at the injection site followed by fatigue were most frequently reported (Fig. 1b). Based on individual perception of adverse events compared

**Table 1 | Demographic and clinical characteristics of the study population**

| | Infected n = 64* | Non-infected n = 63 | p-value |
|---|---|---|---|
| **Years of age [mean ± SD]** | 46.5 ± 13.7 | 52.6 ± 12.7 | 0.011[§] |
| **Sex, n (%)** | | | 0.090[†] |
| male | 26 (40.6) | 16 (25.8) | |
| female | 38 (59.4) | 47 (74.6) | |
| **Vaccine regimen** | | | 0.473[†] |
| mRNA | 40 (62.5) | 35 (55.6) | |
| Vector/mRNA combination | 24 (37.5) | 28 (44.4) | |
| **Number of immunization events (n)[#]** | | | <0.0001[†] |
| 3 | 5 (7.8) | 55 (88.7) | |
| 4 | 59 (92.2)* | 8 (12.7) | |
| **Analysis time [days after vaccination], median (IQR)** | 14 (2) | 15 (4) | 0.109[‡] |
| **Differential blood cell counts [cells/µl], median IQR** | n = 63 | n = 60 | |
| Leukocytes | 6650 (2375) | 7100 (2900) | 0.134[‡] |
| Granulocytes | 4017 (1888) | 4351 (2350) | 0.166[‡] |
| Monocytes | 513 (192) | 573 (317) | 0.008[‡] |
| Lymphocytes | 2011 (766) | 2086 (831) | 0.977[‡] |
| Thrombocytes | 264000 (89000) | 293000 (82000) | 0.147[‡] |
| **Time from infection to vaccination, median (IQR), range** | 206 (49), 60-869 | n.a. | |
| **Dominant SARS-CoV-2 strain at infection prior to vaccination[$]** | | | |
| Parental | 5 (7.7) | | |
| Alpha | 1 (1.6) | | |
| Delta | 2 (3.1) | | |
| Omicron | 52 (80.0) | | |
| BA.1 | 7 | | |
| BA.2 | 41 | | |
| BA.4/5 | 4 | | |
| Unknown* | 4 (6.3) | | |

[§]Unpaired t-test.
[†]Fisher test; [‡]Mann–Whitney test (both two-sided);
[#]immunization events (including infection(s) or vaccinations) prior to bivalent vaccination;
[$]based on dominance of SARS-CoV-2 strain at the time of individual infection;
*includes 4 individuals without known history of infection, but positive NCP-ELISA. Source data are provided as a source data file.

with previous doses, the bivalent vaccine was generally better tolerated than previously received COVID-19 vaccines, with some minor differences between individuals with and without prior infection (Fig. 1c, $p = 0.032$).

### Lower levels of spike-specific IgG and neutralizing activity in non-infected individuals

A schematic overview of blood sampling before and after vaccination is shown in Fig. 2a. Both individuals with and without prior infection had detectable spike-specific IgG before vaccination at baseline with significantly higher median levels in previously infected individuals with hybrid immunity (1714 (IQR 1921) BAU/ml) than in non-infected (465 (IQR 816) BAU/ml, $p < 0.0001$, Fig. 2b). The bivalent vaccine led to a significant increase in IgG-levels in both groups ($p < 0.0001$). Although individuals with hybrid immunity reached significantly higher IgG-levels (7544 (IQR 5566) BAU/ml) than non-infected (5045 (IQR 4751) BAU/ml), the relative increase was significantly higher in individuals without prior infection (9.5-fold versus 4.8-fold, $p < 0.0001$,

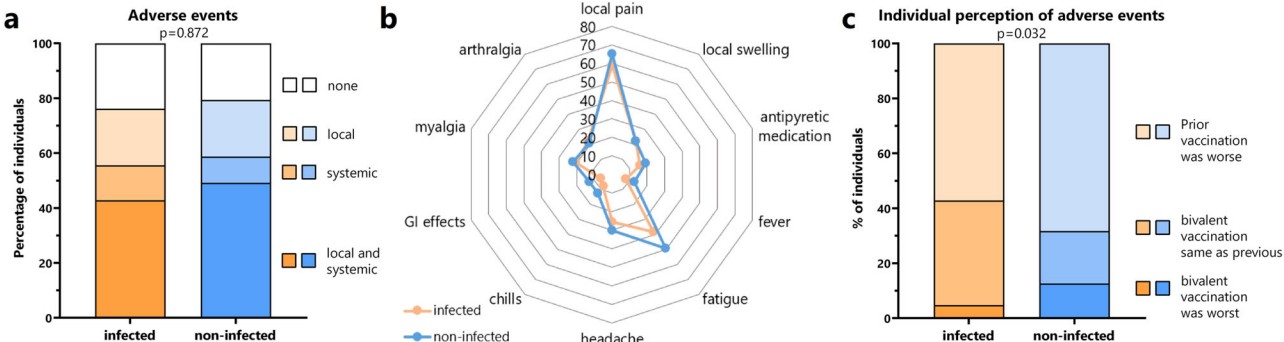

Fig. 1 | Adverse events after the bivalent vaccination. a Local and systemic adverse events in the first seven days after the bivalent vaccination were self-reported using a questionnaire. Shown is the percentage of individuals with ($n = 64$) and without infection ($n = 63$) who reported local or systemic adverse events or both. b The percentage of individuals with individual local or systemic adverse events and use of antipyretic medication. c Individual perception of relative severity of adverse events is shown based on whether individuals had felt more affected by the bivalent dose or by a vaccine dose administered before. Statistical analysis was performed by the $X^2$ test (two-sided). Source data are provided as a source data file.

Fig. 2c). As shown by a micro-neutralization assay, baseline neutralizing activity towards the authentic parental SARS-CoV-2 strain (D614G, FFM7) were significantly higher in previously infected than in non-infected individuals and significantly increased after vaccination in both groups (at least 8 fold, $p < 0.0001$, Fig. 2d). Likewise, median titers after vaccination were also significantly higher in individuals with hybrid immunity ($p < 0.0001$), and more than 50% of infected individuals reached neutralization titers above the upper limit of quantification. Compared to FFM7, baseline neutralizing activity towards the authentic Omicron subvariant BA.5 targeted by the vaccine, as well as BA.1 and BA.2 was generally lower with marked differences between previously infected and non-infected individuals ($p < 0.0001$, Fig. 2d). In line with the fact that most infections had occurred in the BA.2 era (Table 1), individuals with hybrid immunity had higher median baseline neutralizing titers towards BA.2 than towards BA.1 or BA.5. Their vaccine-induced increase in neutralizing titers was more pronounced for BA.5 (8 fold) than for BA.1 or BA.2 (by ≥4 fold). Among non-infected individuals, the majority had negative baseline titers towards all Omicron subvariants, which increased ≥16 fold not only for BA.5, but also for BA.1 and BA.2 (Fig. 2d). As shown by the correlation matrix in Fig. 2e, IgG levels and neutralizing antibody activity towards the various Omicron subvariants showed significant correlations in both individuals with and without previous infection.

## Quantitative similarities in spike-specific CD4 and CD8 T cells towards parental spike and Omicron subvariants irrespective of infection history

Specific CD4 and CD8 T cells towards the parental spike were quantified before and after vaccination. Specific T cells were identified after stimulation with overlapping peptides followed by intracellular staining of IFNγ in CD69 positive CD4 and CD8 T cells with a representative example shown in Fig. S2. As shown in Fig. 3a, the vaccine induced a significant increase in both specific CD4 and CD8 T cells ($p < 0.0001$). Interestingly, unlike spike-specific antibodies, there was no difference between individuals with and without prior infection, which held true for both baseline levels and vaccine-induced levels of specific CD4 and CD8 T cells (Fig. 3a). In contrast, changes in the magnitude of polyclonally stimulated CD4 and CD8 T cells were less pronounced (Fig. 3b).

We also analyzed vaccine-induced CD4 and CD8 T cells against the Omicron BA.4/5 spike targeted by the vaccine, and against the two other Omicron subvariants BA.1 and BA.2. As shown in Fig. 3c, vaccine-induced CD4 and CD8 T cell levels against the Omicron variants were of similar magnitude as those against the parental spike. Moreover, unlike antibodies, specific CD4 and CD8 T cell levels did not differ between individuals with and without prior infection (Fig. 3c). Within either CD4 or CD8 T cells, there was a strong correlation in the percentages of T cells towards parental spike and BA.1, BA.2 and BA.4/5. Interestingly, however, spike-specific CD4 and CD8 T cells only correlated in individuals with previous infection, whereas no such correlation was found in non-infected individuals (Fig. 3d). Disaggregated data of IgG, neutralizing antibodies, CD4 and CD8 T cells for females and males are shown in Figs. S3 and S4, respectively.

## Correlations between spike-specific IgG, neutralizing activity and cellular immunity

We next analyzed correlation patterns of CD4 and CD8 T cells with humoral immune response parameters. When analyzing the whole group of individuals irrespective of prior infection, a correlation between CD4 T cells and IgG as well as neutralizing antibodies towards the parental strain and Omicron subvariant BA.2 and BA.4 was found (Fig. S5). Interestingly, however, when stratified for individuals with and without prior infection correlation patterns of CD4 and CD8 T cells with humoral immune response parameters showed some distinct differences that were unmasked by combined analysis (Fig. 4). In individuals with prior infection, specific CD8 T cell levels towards spike of the parental strain or of the Omicron subvariants BA.1, BA.2 or BA.4/5 correlated with IgG titers, and neutralizing activity towards the parental SARS-CoV-2, and in part towards Omicron BA.1 or BA.2, whereas specific CD4 T cells did not show any correlation with humoral immunity (Fig. 4a). In contrast, among individuals without prior infection, specific CD4 T cell levels towards spike of the parental SARS-CoV-2 or of the Omicron subvariants BA.1, BA.2 or BA.4/5 correlated with IgG titers, and neutralizing activity towards the parental SARS-CoV-2, and BA.5, and in part towards BA.1 or BA.2 variants, whereas specific CD8 T cells did not show any correlation with humoral immunity (Fig. 4b).

## Phenotypical and functional similarities in spike-specific CD4 and CD8 T cells towards parental spike and Omicron subvariants irrespective of infection history

Apart from quantitative analyses, spike-specific CD4 and CD8 T cells were further characterized phenotypically for their expression of CTLA-4, which has been shown to be upregulated on antigen-specific T cells in response to recent antigen encounter during infections or vaccinations[15–19]. CTLA-4 expression on spike-specific CD4 and CD8 T cells significantly increased after vaccination (figure S4). Moreover, CLTA-4 expression on spike-specific T cells after vaccination was significantly higher as compared to Staphylococcus aureus Enterotoxin B (SEB)-reactive CD4 and CD8 T cells (Fig. 5a), which indicates that the

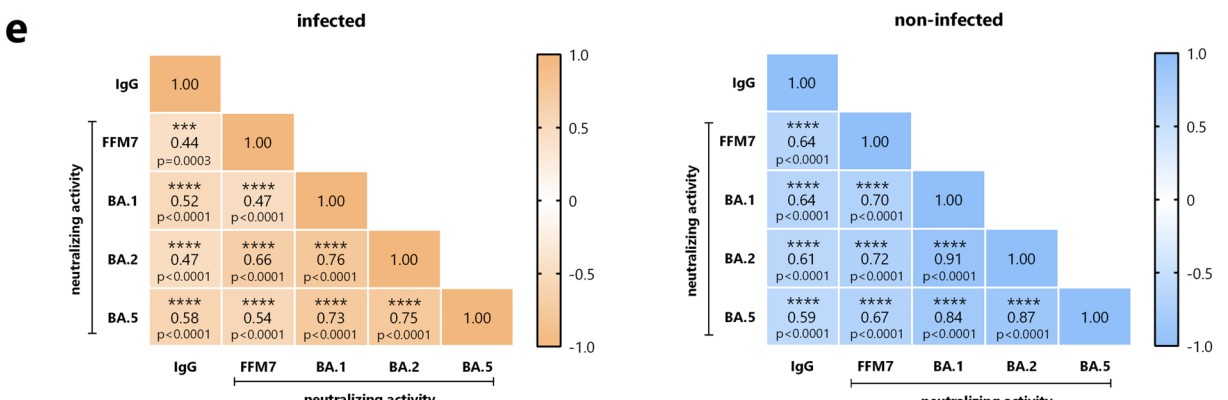

effect is vaccine-specific. Moreover, CTLA-4 expression levels of vaccine-induced CD4 and CD8 T cells reactive towards spike of the parental strain and the Omicron subvariants were similarly high irrespective of prior infection. Functional analysis of spike-specific CD4 and CD8 T cell subpopulations with the ability to produce IFNγ, TNF and IL-2 alone or in combination showed that the cytokine expression profile was similar for cells reactive towards parental strain and the three VOCs (Fig. 5b). Specific CD4 T cells were predominantly multi-functional with the ability to simultaneously produce all tested cytokines, whereas specific CD8 T cells were predominantly producing IFNγ and TNF and largely lacked the ability to express IL-2. As with CTLA-4 expression, there was no difference in the cytokine profiles of CD4 and CD8 T cells between individuals with and without prior infection (Fig. 5b).

**Fig. 2 | Spike-specific IgG and neutralizing antibodies before and after bivalent vaccination. a** Schematic representation of study design and blood sampling. **b** Spike-specific IgG levels (in BAU/ml) towards the parental SARS-CoV-2 spike protein were determined from individuals with (orange symbols, $n = 64$) and without prior infection (blue symbols, $n = 63$) before and after bivalent vaccination. Statistical analysis (two-sided) was performed using the paired $t$-test (before/after) or the non-parametric Mann-Whitney test for between-group comparisons at baseline and after vaccination. **c** The fold increase in spike-specific IgG levels was determined for individuals with ($n = 64$) and without prior infection ($n = 63$) and compared using Mann-Whitney test (two-sided). **d** Neutralizing activity of antibodies towards authentic parental SARS-CoV-2 (FFM7) and Omicron subvariants

were determined in infected ($n = 63$) and non-infected individuals ($n = 63$) using a microneutralization assay, and differences were calculated using the paired $t$-test (before/after) or the non-parametric Mann-Whitney test (both two-sided) for between-group comparisons at baseline and after vaccination. **e** Correlation matrix between IgG levels and neutralizing activities towards Omicron subvariants. Correlation coefficients were calculated according to two-tailed Spearman and displayed using a color code, and $p$-values (including stars denoting levels of statistical significance) are indicated. Lines or bars in panels (**a–d**) indicate medians and interquartile ranges. SEB, *Staphylococcus aureus* Enterotoxin B. Numbers refer to biologically independent samples examined in one experiment per individual per time point. Source data are provided as a source data file.

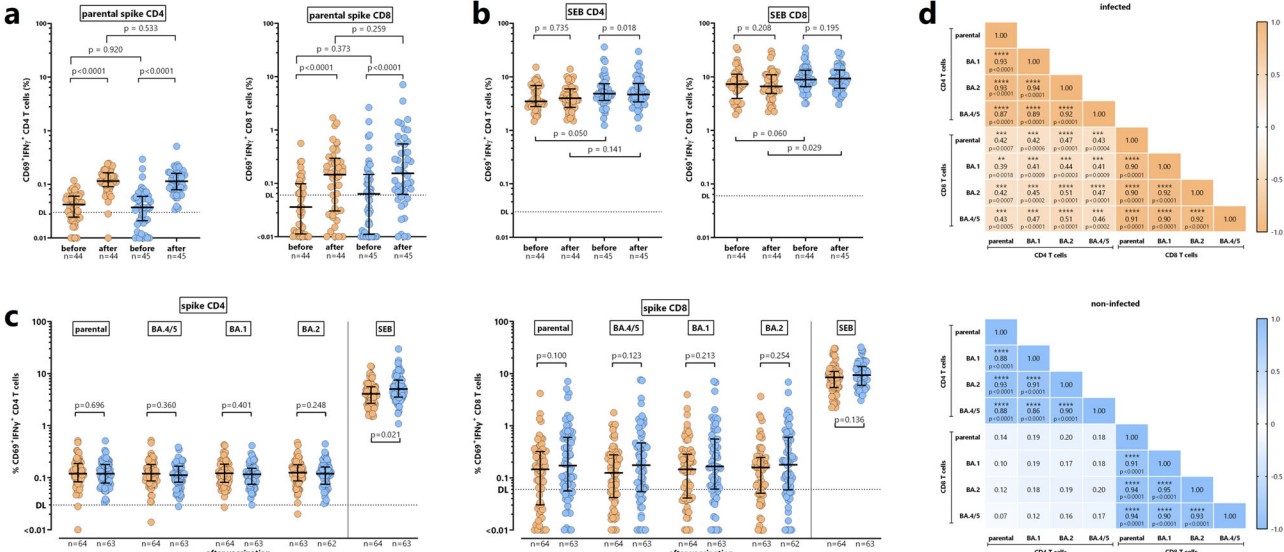

**Fig. 3 | Spike-specific CD4 and CD8 T cells towards parental spike and Omicron subvariants. a** Specific CD4 and CD8 T cells towards SARS-CoV-2 parental spike and (**b**) SEB-reactive CD4 and CD8 T cells were quantified in subgroups of infected (orange symbols, $n = 44$) and non-infected individuals (blue symbols, $n = 45$) before and after bivalent vaccination. Reactive CD4 and CD8 T cells were quantified after stimulation based on expression of CD69 and IFNγ. Statistical analysis was performed using the Wilcoxon matched pairs $t$-test (before/after) or the Mann-Whitney test (both two-sided) for between-group comparisons at baseline and after vaccination. **c** CD4 and CD8 T cells towards parental spike and towards the spike-protein of the Omicron subvariants BA.4/5, BA.1 and BA.2 were determined after bivalent vaccination (infected $n = 64$ ($n = 63$ for BA.2), non-infected $n = 63$). SEB-

reactive CD4 and CD8 T cell levels were quantified as positive controls. Statistical analysis was performed using the Mann-Whitney test (two-sided). Stippled lines denote detection limits (0.03% for CD4 T cells and 0.06% for CD8 T cells). **d** Correlation matrix of specific CD4 and CD8 T cells towards the parental spike and spike of the Omicron subvariants in infected and non-infected individuals. Correlation coefficients were calculated according to two-tailed Spearman and displayed using a color code, and $p$-values (including stars denoting levels of statistical significance) are indicated. Lines in panels (**a–c**) indicate medians and interquartile ranges. SEB, *Staphylococcus aureus* Enterotoxin B. Numbers refer to biologically independent samples examined in one experiment per individual per time point. Source data are provided as a source data file.

## Lower vaccine-induced neutralizing activity, and spike-specific CD4 T cell levels in non-infected individuals with subsequent breakthrough infection

Both individuals with and without prior infection were followed up for development of a breakthrough infection after a median observation time of 146 (IQR 10) days using a questionnaire (Fig. 6a). 25/126 (19.8%) individuals developed a breakthrough infection at a median of 129 (IQR 65) days after the bivalent vaccination, which occurred more often in individuals without prior infection (16/62 (25.8%)) than in individuals with a previous infection (9/64 (14.1%), $p = 0.045$, Fig. 6b and table S1). The incidence rates did not differ significantly (106 (95% CI 49-202) cases/100.000 person-days and 194 (95% CI 111-315) cases/100.000 person-days in individuals with and without prior infection, $p = 0.145$). Most individuals only had mild or moderate symptoms, none of which required hospitalization. Although the duration of these symptoms did not differ between individuals with and without prior infection, symptoms lasting 7 days or more were numerically more frequent among individuals without previous infection (table S1).

We then analyzed whether the levels of humoral and cellular immunity determined prior to vaccination (Fig. S7) or two weeks after

vaccination differed in individuals with and without subsequent breakthrough infection. Data on vaccine-induced immunological parameters for the whole population of study participants are shown in Fig. S8, and results for the subgroups of previously infected and non-infected individuals are shown in Fig. 6. Vaccine-induced IgG levels did not differ in individuals with and without breakthrough infections (Fig. S8c and 6c). In the whole population, vaccine-induced neutralizing antibodies were lower in persons with subsequent breakthrough infection (Fig. S8d, with odds ratios of 0.65-075 depending on the viral strain, table S2). In contrast, spike-specific CD4 or CD8 T cell levels were not statistically different (figure S8e-f), and did not improve test characteristics (area under the curve, AUC) when added to a predictive model (Table S3).

We then performed separate analyses for individuals with and without prior infection (Fig. 6, Tables S2 and S3). Interestingly, both vaccine-induced neutralizing activity and spike-specific CD4 T cell levels towards the parental strain and all Omicron VOCs were significantly lower in previously non-infected individuals who developed a breakthrough infection, whereas no such difference was found for spike-specific CD8 T cells (Fig. 6d–f, right panels). As shown from

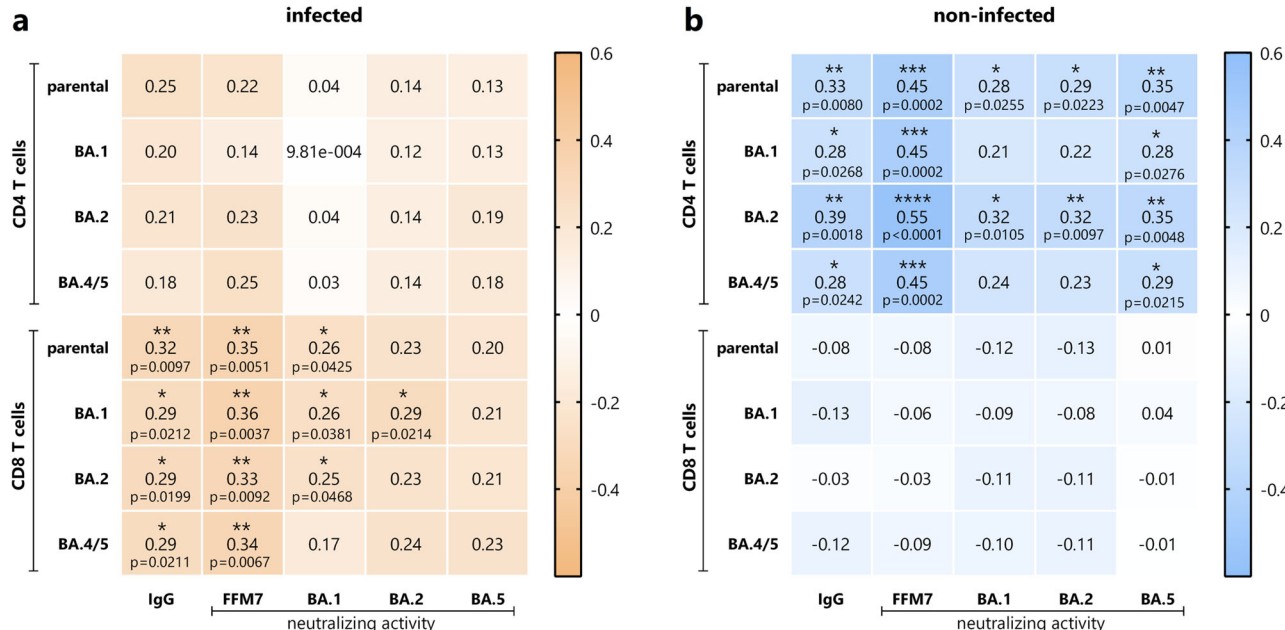

**Fig. 4 | Correlation between vaccine-induced cellular and humoral immunity.** Correlation matrix between vaccine-induced spike-specific CD4 or CD8 T cells with IgG levels and neutralizing activities towards parental SARS-CoV2 and Omicron subvariants in individuals (**a**) with (*n* = 63) and (**b**) without prior infection (*n* = 62). Correlation coefficients were calculated according to two-tailed Spearman and displayed using a color code, and *p*-values (including stars denoting levels of statistical significance) are indicated. Numbers refer to biologically independent samples examined in one experiment per individual per time point. Source data underlying theses analyses are provided as a source data file.

multivariate logistic regression analyses in each subgroup, CD4 T cells and neutralizing antibodies had independent predictive value for breakthrough infections among previously non-infected, with some differences regarding reactivity towards the parental strain and Omicron subvariants (Table S2). Moreover, test characteristics (AUC) among previously non-infected individuals based on neutralizing antibodies increased when CD4 T cells were added to the model (Table S3). In contrast, vaccine-induced neutralizing antibody activity and spike-specific CD4 or CD8 T cells did not differ among previously infected individuals with or without subsequent infection (Fig. 6d–f, left panels, and Tables S2 and S3).

After separate analysis of the two groups, we finally performed an interaction analysis to test for evidence of different immunological associations with protection between the previously infected and non-infected groups. Although this analysis is limited by sample size[20,21], we extended our multiple regression analysis of the whole cohort and added prior infection status as an interaction term on the predictors. As shown in Table S4, there was a statistically significant interaction between prior infection status and parental and Omicron BA.4/5-specific CD4 T cell levels, whereas the interaction between prior infection and other predictors did not reach statistical significance.

## Discussion

In this observational study we show that the bivalent Comirnaty Original/Omicron BA.4/5 vaccine strongly induced specific IgG levels, neutralizing activity, and specific CD4 and CD8 T cells, which were not only directed towards the spike proteins of the parental SARS-CoV-2 and BA.4/5 strains targeted by the vaccine, but also towards Omicron subvariants BA.1 and BA.2. Approximately 50% of our study participants had hybrid immunity based on prior infection, and the vaccine was well tolerated in both groups. However, individuals with and without hybrid immunity showed marked differences in the induction of both IgG levels and neutralizing antibody titers, whereas vaccine-induced T cell levels were induced to a similar extent. Finally, we show that individuals without prior infection more frequently developed a

SARS-CoV-2 breakthrough infection after vaccination. We also found that previously non-infected individuals who subsequently developed breakthrough infections not only had significantly lower neutralizing activity, but also lower spike-specific CD4 T cell levels after vaccination, whereas no such differences were found among previously infected individuals.

Vaccine-induced IgG levels and neutralizing antibody activity towards the various Omicron subvariants showed significant correlations in both individuals with and without previous infection. Higher antibody levels and neutralizing activity towards the ancestral spike as compared to the Omicron subvariants indicate some extent of immune imprinting and are in line with other reports on neutralizing antibodies after vaccination with a bivalent BA.1[6] or BA.4/5 vaccine[7–10]. However, these studies were of small sample size[7–10,22], did not differentiate between CD4 and CD8 T cells[22], reported aggregated data for vaccinated individuals with and without prior infection[9,10] or did not report pre-vaccination data[8,10] to appreciate dynamic changes in neutralizing activity before and after vaccination in individuals with and without prior infection. In general, immune imprinting by previous exposures with the monovalent vaccine may account for the relative dominance of neutralizing activity towards the parental spike as compared to the Omicron subvariants[23,24]. Among individuals with prior infection, it was interesting to note that baseline immunity against the ancestral SARS-CoV-2 and the BA.2 variant was higher than against BA.1 or BA.5. This well reflects previous exposure to these antigens by vaccination and subsequent BA.2 infection, which was the most prevalent infection strain in our study participants. Despite some extent of imprinting, the most pronounced increase was found for antibody titers towards the two SARS-CoV-2 spike variants against which the vaccine was also directed. Moreover, consistent with infection representing one more immunization event, individuals with prior infection had both higher baseline titers and reached higher absolute levels of vaccine-induced humoral immunity. As with other vaccines such as influenza[25], higher levels of baseline antibody titers were associated with a less pronounced relative increase after vaccination.

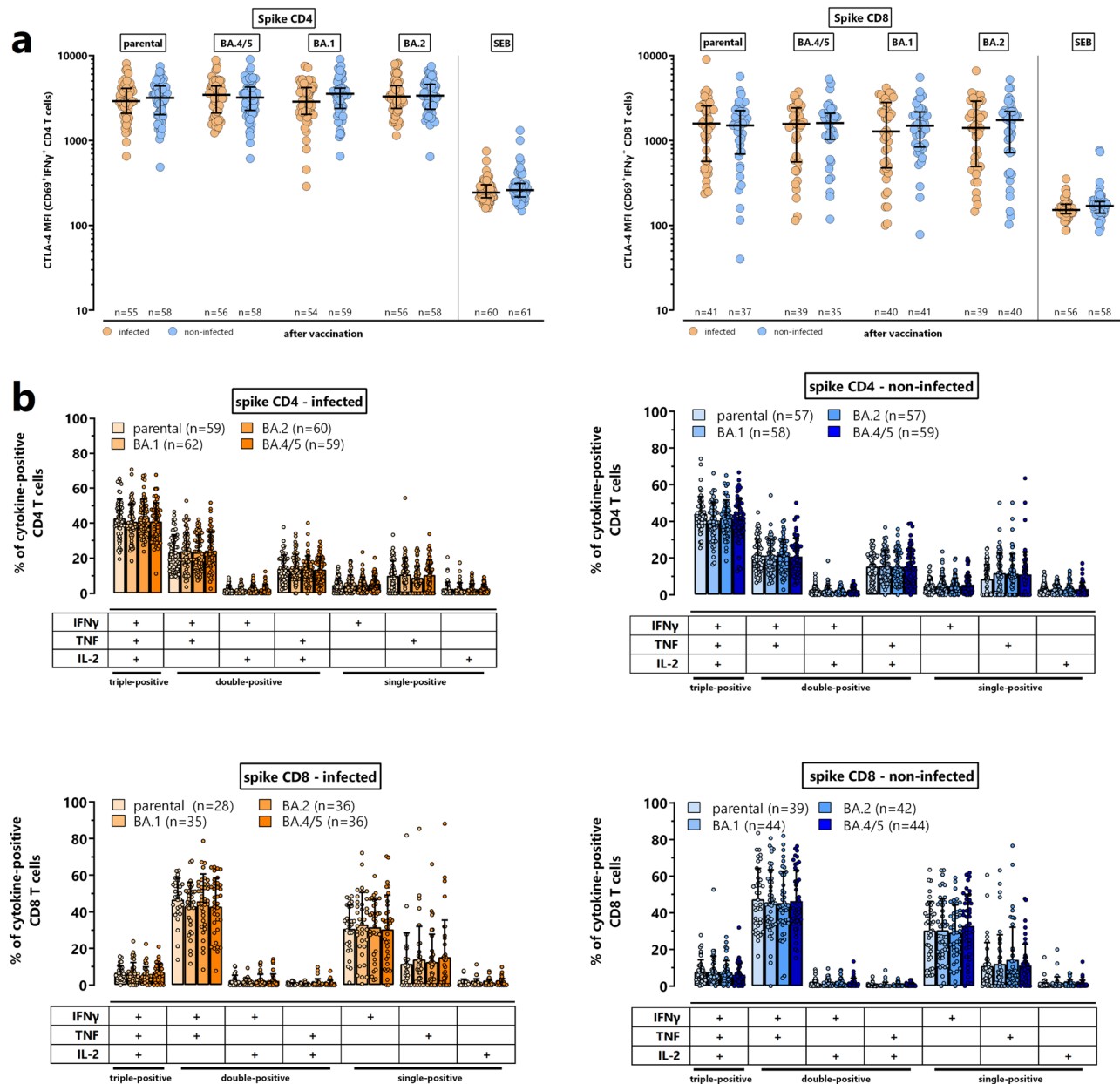

**Fig. 5 | CTLA-4 expression and cytokine profile of spike-specific and SEB-reactive CD4 and CD8 T cells. a** Specific CD4 and CD8 T cells towards the parental spike and Omicron subvariants BA.4/5, BA.1 and BA.2 as well as SEB-reactive CD4 and CD8 T cells were analyzed for expression of CTLA-4, which is expressed as median fluorescence intensity (MFI). All samples (from all individuals) were analyzed, but to ensure robust statistics, this analysis was restricted to samples with at least 20 CD69 + IFNγ + CD4 or CD8 T cells (with sample size indicated in the figures). Lines represent medians with interquartile ranges. Differences between the groups were calculated using the Mann Whitney test (two-sided). **b** Specific CD4 and CD8 T cells towards the parental spike and Omicron subvariants BA.4/5, BA.1 and BA.2 were analyzed for expression of IFNγ, TNF and IL-2 alone or in combination after Boolean gating. This allowed distinction of seven subpopulations expressing three, two or a single cytokine. All samples (from all individuals) were analyzed, but to ensure robust statistics, only samples with at least 30 cytokine-expressing spike-specific CD4 or CD8 T cells after normalization to the negative control stimulation were considered (with sample size indicated in the figures). Bars represent means and standard deviations, and differences between the groups were analyzed using the Kruskal-Wallis with Dunn's post test. CTLA-4, cytotoxic T-lymphocyte antigen 4; SEB, *Staphylococcus aureus* Enterotoxin B. Numbers refer to biologically independent samples examined in one experiment per individual. Source data are provided as a source data file.

In this regard, the relative increase in vaccine-induced humoral immunity was more pronounced in individuals without prior infection, which was particularly striking for neutralizing titers towards the Omicron subvariants, which were negative in most cases at baseline and increased 16 fold after vaccination. It therefore seems that the additional bivalent vaccine dose may confer a stronger benefit for previously non-infected individuals.

Unlike antibody levels, baseline frequencies of spike-specific CD4 or CD8 T cells were equally low in individuals with and without prior infection and both groups showed a similar increase after bivalent vaccination. In general, vaccine-induced T cells were largely poly-functional with high CTLA-4 expression levels as sign of recent antigen encounter, which is known to increase on antigen-specific T cells during vaccinations or active infections, and decrease thereafter[15–19]. Moreover, CD8 T cell levels were higher than CD4 T cells. Interestingly, spike-specific CD4 and CD8 T cell levels only correlated in individuals with previous infections, whereas no such correlation was observed among non-infected individuals. This may suggest that the potent

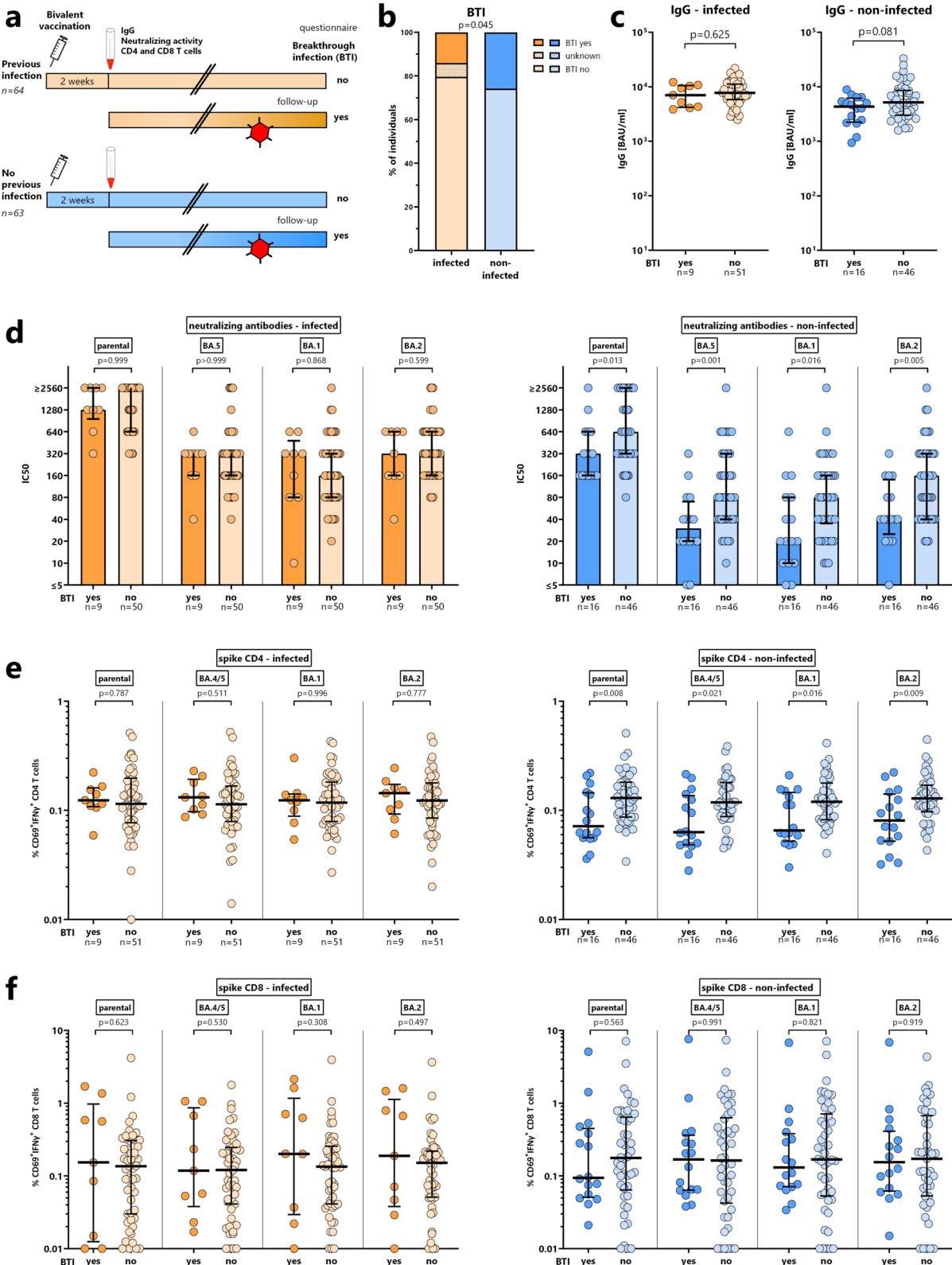

**Fig. 6 | Vaccine-induced humoral and cellular immunity in individuals with and without breakthrough infections. a** Schematic outline of the study design. All study participants were followed up until March 2023 for development of breakthrough infections based on self-reporting using a questionnaire. Immune parameters in the following panels refer to results two weeks after the bivalent vaccination. **b** Percentage of breakthrough infections (BTI) in individuals with (orange, *n* = 64) and without (blue, *n* = 62) prior infection. Bivalent vaccine-induced spike-specific (**c**) IgG levels, (**d**) neutralizing antibody activity, (**e**) CD4 T cells, and (**f**) CD8 T cells in individuals with and without prior infection, stratified for individuals with and without subsequent breakthrough infection. Statistical analysis was performed using the Mann-Whitney test (two-sided). Lines or bars in panels (**c**–**f**) indicate medians and interquartile ranges. BTI, breakthrough infection. Numbers refer to biologically independent samples examined in one experiment per individual. Source data are provided as a source data file.

induction of both CD4 and CD8 T cells by natural infection[26] may allow for a more uniform expansion of both T cell populations after re-challenge with the vaccine. Within each population of CD4 or CD8 T cells, a strong correlation and striking similarity of vaccine-induced T cell levels towards the parental spike and spike of all tested Omicron subvariants was found. This is in line with results after monovalent vaccination[27,28], and indicates substantial cross-reactivity of T cells with little evidence of immune escape. In general, evolution of immune escape mutants affecting T cells is less frequent due to the diversity of MHC alleles in the human population. Cross-reactivity on the T cell level may on one hand result from the fact that more than 86% of class I and 72% of class II epitopes of the Omicron spike are fully conserved, and epitope mutations do not necessarily preclude T cell recognition[27]. On the other hand, reactive T cells have been shown to be directed towards non-mutated regions of the spike protein[29]. Data on the induction of T cells after bivalent BA.4/5 vaccination are scarce. The marked increase in vaccine-induced T cell levels in our study cohort contrasts with findings among 18 individuals where the bivalent vaccine did not lead to a substantial augmentation in cellular immunity[9]. The reason for this difference is unclear but it seems that T cell levels at baseline were higher than in our study, which may be due to a smaller interval between the previous immunization event and the bivalent vaccination and/or differences in infecting strains. As the authors speculate that the majority of their participants had hybrid immunity due to a recent BA.5 infection during the summer and fall of 2022[9], this may have boosted specific T cell immunity already prior to vaccination thereby diminishing a further booster effect by the vaccine.

Data on the effectiveness of bivalent vaccines in individuals with and without prior infection are limited, but previous studies have shown that hybrid immunity conferred stronger protection from Omicron infection than monovalent vaccine-induced immunity alone[30–33]. Early estimates of bivalent vaccines in general suggest a high effectiveness against hospitalization and death when given as additional dose or as alternative to the monovalent dose[34–36]. As with a recent study after bivalent BA.1/2 vaccination[6], our study after bivalent BA.4/5 vaccination also showed a lower percentage of breakthrough infections among individuals with a previous infection, which mostly occurred during the BA.2 wave in Germany. So far, data linking results on bivalent vaccine-induced humoral and cellular immunity to subsequent breakthrough infections are lacking. Although the number of breakthrough infections among previously infected individuals was low, our data indicate that vaccine-induced humoral and cellular immunity among previously infected individuals did not differ between individuals with and without subsequent infection, which should be confirmed in larger cohorts of individuals with a history of infection. Our observations may result from the fact that previously infected individuals had higher levels of IgG and neutralizing activity as compared to previously non-infected, and may indicate that other components of the adaptive immune response such as local immunity in the respiratory tract may play an additional role towards protection[37]. Interestingly, among previously non-infected individuals, those with subsequent breakthrough infections had significantly lower levels of vaccine-induced neutralizing activity and CD4 T cells. This is conceivable, as helper CD4 T cells are instrumental in supporting induction of antibodies, and both parameters showed a significant correlation in non-infected individuals only. In line with our results, previously non-infected renal transplant recipients after a third dose of monovalent vaccination showed a similar association of low levels of neutralizing antibodies and T cells with subsequent breakthrough infection[38]. However, as patients with prior infection were excluded, this study did not address the effects of prior infection on vaccine-induced immunity and subsequent infections. Neutralizing antibodies were already characterized as the promising correlate of protection mostly derived from efficacy studies with large sample sizes, whereas T cells were not concomitantly studied on a larger scale due to a higher

technical complexity[32,39–42]. Low neutralizing antibody levels were also identified to be predictive of both infection and severity of disease in a prospective household study among contacts without previous infections during the Delta wave[43]. T cell activity was also determined by an ELISPOT assay and did not have any predictive value, but given that CD4 and CD8 T cells were not distinguished, this may have masked a protective effect of individual T cell subpopulations. As shown from our data, neutralizing antibodies and CD4 T cell levels towards most strains correlated among previously non-infected individuals and low levels were independent indicators for subsequent breakthrough infection. Future studies are needed to further evaluate the role of vaccine-induced neutralizing antibodies and CD4 T cells as indicator of protection in individuals with and without prior infection.

Our study has some limitations. Analyses of breakthrough infection relied on self-reporting which may have underestimated the true infection rate, and we were unable to address correlations with severe disease as none of the breakthrough infections required hospitalization. Moreover, we have not performed follow-up analyses to assess peri-infection immune responses. Nevertheless, together with the known waning of humoral immunity following vaccination[44–46], lower post-vaccine responses may have more rapidly decreased beyond a level of protection on follow-up. Due to the real-world setting, we were unable to study effects of a monovalent booster dose, as the bivalent vaccination was preferentially recommended at the start of our study. A strength of our study is the considerably large sample size with similar numbers of individuals with and without history of prior infection, which allows evaluating the effect of prior infection on immunogenicity. All vaccinations were carried out in the same time frame in the same region with similar circulating variants before and after vaccination. Moreover, we performed analyses of both humoral and cellular immune responses including CD8 and CD4 T cells, although our analysis was restricted to Th1 cytokines and did not include further subpopulations such as follicular helper T cells. Finally, the considerably high infection rate in the Omicron BA.5 wave of around 20% allowed description of individual participant-based immunological indicators of protection. Nevertheless, overall sample size was too low to perform meaningful analysis of cut-offs as a correlate of protection, and we cannot be certain whether the lack of a statistical significance in the interaction analysis of prior infection with immunological predictors other than CD4 T cells is due to no effect or due to limited sample size.

In conclusion, we have shown that the bivalent BA.4/5 vaccine was well tolerated and was strongly immunogenic with marked difference in individuals with and without prior infection. We found a correlation between bivalent vaccine-induced neutralizing antibodies and CD4 T cells among individuals without prior infection, and their levels were lower in those with subsequent breakthrough infection. Our results on the immunogenicity of vaccinations in previously infected individuals may be of relevance given the ongoing SARS-CoV-2 infection waves, where most individuals will soon have a history of one or more infections. This will facilitate future studies with larger sample size to further address the lack of correlation between vaccine-induced neutralizing antibodies and CD4 T cells among previously infected individuals, and implications for further breakthrough infections.

## Methods
### Ethical regulations
The study was approved by the ethics committee of the "Ärztekammer des Saarlandes" (reference 76/20 including amendment), and all individuals gave written informed consent. Participants did not receive any compensation.

### Study participants and study design
In an observational study, immunocompetent individuals receiving a bivalent COVID-19 vaccine (Comirnaty Original/Omicron BA.4-5,

BioNTech/Pfizer) were enrolled in the study between 28th of September and 14th of December 2022 as per German recommendations. Participants were recruited either from the Saarland University Medical Center (Homburg, Germany) or from a public vaccination campaign (St. Ingbert, Germany). A heparinized blood sample was scheduled before and 13–18 days after vaccination to determine differential blood counts and immunogenicity. In addition, information on age and sex (self-reported) was collected, and all participants reported their history of SARS-CoV-2 infection and COVID-19 vaccination, and vaccine-related adverse events in the first week after vaccination, using a standardized questionnaire. Finally, all individuals were interrogated in March 2023 for potential development of a breakthrough infection after the bivalent vaccination (see figure S1).

### Viral strains

The following SARS-CoV-2 isolates were used in this study: Parental (SARS-CoV-2 B.1 FFM7/2020, GenBank ID MT358643), BA.1 (SARS-CoV-2 B.1.1.529 FFM-SIM0550/2021 (EPI_ISL_6959871), GenBank ID OL800702), BA.2 (SARS-CoV-2 BA.2 FFM-BA.2-3833/2022, GenBank ID OM617939), BA.5 (SARS-CoV-2 BA.5 FFM-BA.5-501/2022, GenBank ID OP062267)[3,47–50].

### Quantitation of SARS-CoV-2 specific CD4 and CD8 T cells

SARS-CoV-2 spike-specific T cells were measured after a 6 h stimulation of heparinized whole blood with overlapping peptide pools derived from the S1 and S2 domain of the parental SARS-CoV-2 spike protein (N-terminal receptor binding domain and C-terminal portion including the transmembrane domain, each peptide 2 µg/ml) exactly as previously described[51,52]. In addition, peptide pools from the BA.1, BA.2 and BA.4/5 spike protein were used (jpt, Berlin, Germany, product codes PM-WCPV-S-1, PM-SARS2-SMUT08-1, PM-SARS2-SMUT09-1, PM-SARS2-SMUT10-1, each peptide 2 µg/ml). Stimulations with 0.64% DMSO and with 2.5 µg/ml of *Staphylococcus aureus* enterotoxin B (SEB; Sigma) served as negative and positive controls, respectively. All stimulations were carried out in presence of co-stimulatory antibodies against CD28 and CD49d (clone L293 and clone 9F10, 1 µg/ml each), and 10 µg/ml brefeldin A was added after 2 h of stimulation. After a total of 6 h, cells were treated with 20 mM EDTA for 15 min and fixed using BD lysing solution based on the manufacturer´s instruction (BD). Thereafter, cells were washed with FACS-buffer (PBS, 5% filtered FCS, 0.5% bovine serum albumin, 0.07% NaN$_3$), and immunostaining was performed using anti-CD4 (clone SK3, 1:33.3), anti-CD8 (clone SK1, 1:12.5), anti-CD69 (clone L78, 1:33.3), anti-IFNγ (clone 4 S.B3, 1:100), anti-IL-2 (clone MQ1-17H12, 1:12.5), anti-TNF (clone MAb11, 1:20) and anti-CTLA-4 (clone BNI1, 1:50). Antigen-specific T cells were compared on a quantitative and functional basis. Spike-specific CD4 or CD8 T cells were identified by co-expression of CD69 and IFNγ and further characterized for expression of CTLA-4, and of the cytokines IL-2 and TNF. Specific CD4 or CD8 T cell levels were determined after subtraction of control stimulations and cut-offs displayed in the figures were based on the distribution of the negative control reactivity (≥ 0.03% for CD4 and ≥0.06% for CD8 T cells, respectively). Although CD69-negative T cells rarely produce cytokines, both co-staining of CD69 as early antigen-specific activation marker in our staining panel as well as subtracting the negative control from the specific stimulation was chosen as measures to increase specificity. Analyses were carried out on a FACS Canto II using Diva software (BD, Heidelberg, Germany). The gating strategy is shown in figure S9, and further information on antibodies is given in table S5.

### Quantitation of SARS-CoV-2 specific IgG and neutralizing activity

The amount of SARS-CoV-2 spike-specific IgG antibodies was determined using an ELISA (SARS-CoV-2-QuantiVac, Euroimmun, Lübeck,

Germany, product code EI 2606-9601-10 G) based on the manufacturer´s instructions as described before[51,52]. Antibody binding units (BAU/ml) <25.6 were scored negative, ≥25.6 and <35.2 were scored intermediate, and ≥35.2 were scored positive. SARS-CoV-2 specific IgG towards the nucleocapsid protein (NCP) were quantified using the anti-SARS-CoV-2-NCP-ELISA based on the manufacturer´s instructions (Euroimmun, product code EI 2606-9601-2 G). The in vitro neutralizing activity of the antibodies was measured using a micro neutralization assay with A549-AT cells[53] and authentic parental SARS-CoV-2 (FFM7, D614G) and the Omicron subvariants BA.1, BA.2, and BA.5. As described previously[3,28], cells were infected with serially diluted sera (1:2) pre-incubated with 4000 TCID$_{50}$/mL of each SARS-CoV-2 variant. Infected cells were monitored for cytopathic effect (CPE) formation 48 h post inoculation.

### Statistical analysis

The Mann-Whitney test or the paired *t*-test was used to analyze differences between non-parametric data such as blood cell populations, T cell and antibody levels, and CTLA-4 expression. The Kruskal-Wallis test was performed for paired analyses of the cytokine-profiles of CD4 and CD8 T cells towards parental spike and spike of the Omicron subvariants. Age was analyzed using a non-paired *t*-test. Categorial analyses were performed using the Fisher´s test or $X^2$ tests. Correlations between the immunological parameters were analyzed using a correlation matrix according to Spearman. Multivariate logistic regression analyses were carried out to identify immunological factors associated with risk for breakthrough infections, including variance inflation factor (VIF) analysis to avoid multicollinearity. We then built the same multivariable regression model that included both main effects and interaction terms. We included two interaction terms that tested interaction between immunological factors and prior COVID-19 infection with breakthrough infection as an outcome. A *p*-value < 0.05 was considered statistically significant. Analyses was carried out using GraphPad Prism 10.0.3 software using two-tailed tests (GraphPad, San Diego, CA, USA).

### Reporting summary

Further information on research design is available in the Nature Portfolio Reporting Summary linked to this article.

## Data availability

Table 1, Figs. 1–6, Tables S1-S4, and Figures S3-S8 have associated raw data and data are available in a public repository (https://zenodo.org/records/10815331). All other data are available in the article and its Supplementary files or from the corresponding author upon request. As age may be subject to confidentiality, data in the repository refer to age groups. Source data are provided with this paper.

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

## Acknowledgements

The authors thank all participants to this study, and the team of the occupational health care center at Saarland University Medical Center and the CDU Rentrisch for their support in enrolling study participants. Expert technical assistance by Christiane Pallas is acknowledged. Financial support was provided in part by the State chancellery of the Saarland to M.S. Moreover, the work was in part supported by the cluster project ENABLE, the Innovation Center TheraNova, and the LOEWE Priority Program CoroPan funded by the Hessian Ministry for Science and the Arts (HMWK) to M.W.

## Author contributions

R.U., T.S. and M.S. designed the study; R.U., S.B., M.W., T.S. and M.S. designed the experiments, R.U., S.B., V.K., S.M., F.H., A.A.-O., M.W. and C.G. performed experiments; A.A.-O., S.S., C. B., S.L.B., B.C.G. and U.S. contributed to study design, patient recruitment, and clinical data acquisition. R.U., S.B., T.S., L.M. and M.S. performed statistical analysis. R.U., T.S., U.S., M.W. and M.S. supervised all parts of the study, and performed analyses; M.S. wrote the manuscript. All authors approved the final version of the manuscript.

## Funding

## Competing interests

M.S. has received grant support from Astellas and Biotest to the organization Saarland University outside the submitted work, and honoraria for lectures from Biotest and Novartis, and for advisory boards from Moderna, Biotest, MSD and Takeda. B.C.G. has received honoraria for lectures from BioNTech, Sanofi, CSL Seqirus, and GSK. M.W. has received research support from Roche, Qiagen outside the submitted work and a speaker's fee from Astra Zeneca. All other authors of this manuscript have no conflicts of interest to disclose.
