## [Peer Review File · Nature Communications]

SARS-CoV-2-specific cellular and humoral immunity after bivalent BA.4/5 COVID-19-vaccination in previously infected and non-infected individualsREVIEWER COMMENTS

Reviewer #1 (Remarks to the Author):

Urschel et al report on antibody titres and CD4/CD8 T cell responses following vaccination with BA4/5 bivalent mRNA booster vaccines. Among the cohort, the group with prior COVID-19 exhibited higher antibody titres both before and after vaccination compared with uninfected participants. Spike-specific T cell frequencies were boosted by vaccination but were relatively similar regardless of infection history. Over the course of 5 months of follow-up, breakthrough infections were observed among both groups. There was no identified correlate of protection among individuals with prior COVID, but the uninfected group exhibited low post-vaccination neutralising antibody titres and CD4 T cell responses in individuals with subsequent breakthrough infection.

Overall, these observations are consistent with other reports in the literature, reflecting the immunogenicity of the bivalent booster vaccines and the ability of S-specific T cells to recognise both ancestral and omicron-related spike proteins.

Can the authors please provide further explanation and references to justify the use of CTLA-4 as a marker of recent antigen exposure (line 157)? As the antigen-specific cells have been re-stimulated in vitro, can it be shown that CTLA-4 expression has not been induced by this antigen exposure? How long does CTLA-4 expression persist after antigen exposure (i.e. what timeframe does 'recent' reflect)?

The longitudinal tracking of breakthrough infection following vaccination is interesting and provides a good opportunity to prospectively assess potential protective immune responses. While the observation that individuals experiencing infection tended to have lower neutralising antibody responses after vaccination, it would be useful to try to understand the factors that underpin these outcomes. Did the breakthrough individuals have particularly poor fold-change responses to vaccination? Did they have low pre-vaccine titres? Was the post-vaccination breadth of those responses less compared to the non-breakthrough cohort? Similarly, did they exhibit differences in their pre-vaccine CD4 T cell responses, or have relatively poor cTFH responses?

Reviewer #2 (Remarks to the Author):

The manuscript by Urschel et al. seeks to determine the impact of bivalent mRNA vaccines in individuals previously infected or not with SARS-CoV-2. The authors found a higher vaccine-induced IgG response in individuals with prior infection compared to non-infected. In contrast, a lower difference was shown in CD4 and CD8 responses. Breakthrough infections were more frequent in individuals with no previous infection who had low humoral and cellular immunity. The data looks interesting and bring the importance of vaccine-induced cellular immunity on subsequent infection and as a possible correlate of protection. There are minor points to be addressed.

Line 80 – It would be interesting to add the interval between the infection to the bivalent vaccination (in the “infected group”) as it can influence the rapid response 2 weeks after vaccination.

Line 141 - Have the authors tried to correlate CD8 and CD4 responses out of activated cells (i.e., no CD69+)? Even though gating on activated cells string the analysis, I’m questioning if you could miss some specific cytokines as the frequency of both CD4 and CD8 is low.

Line 152 – The use of CD40L instead of CD69 would better represent the correlation between IgG and CD4s.

Line 165 – Have the authors tried to analyze the percentage of individuals who are producing 1, 2, or 3 cytokines? This data could be interesting to improve the functional potency of the T cells.

mRNA induces a Th1 profile preferentially, but considering the heterogeneous of the individuals, it also is interesting to analyze Th2 cytokines

Line 345 – How was determine the cut-off of CD4 and CD8 (0.03%)? Was it the same for both?

Line 38-39 – The wording correlation of protection in the aspect of the magnitude of CD4s is “risky”. It would be better to use “an indicator”.

Figure 5 – It would be great to have the N of each group indicated in the graphs (for example: right below the dots). The visualization of sample size would be more informative in statistics analysis.

On extended Figure 3, IgG was correlated with CD4 in infected but was not in uninfected.

Have the authors tried to perform this analysis with the “no BTI” group to see if the CD4 and IgG had a coordinated response towards protection?

The percentage of BTI was more frequent in the non-infected group and in those who had low humoral and cellular immunity two weeks after bivalent mRNA vaccination. The authors have mentioned that the magnitude of vaccine-induced neutralizing activity and specific CD4 T cells may serve as a correlate for protection. However, no further analysis with the indicators was performed – how are the magnitude reference value for protection (CD4 frequency, IgG titer/activity, etc)? Additionally, the sample size for “BTI yes” seems too low to determine that. I would use “an indicator of protection” instead.

Reviewer #3 (Remarks to the Author):

The authors present a well written study of a cohort of individuals (n=127) who received an Omicron BA.4/5 bivalent booster vaccine. The authors present assessments of immunogenicity and compare these between those who had previous infection (before the boost) and those who did not report having prior infection (and had N-negative serology). Unsurprisingly they find that those with previous infection had higher neutralising antibody titers (but a smaller boost in titer) – consistent with other reports. They then examine breakthrough infections in these cohorts and report associations between neutralizing antibodies and CD4 T cells and risk of breakthrough infections (but only in the subset of those not previously infected). From this the authors draw somewhat problematic conclusions about “correlates of protection”. Unfortunately, given the number of other very similar studies, and problematic conclusions around correlates, this study presents a fairly incremental advance on the existing work in the field.

1) Specifically, this reviewer’s greatest concern is that the unwary reader may understand the authors as having shown that T cells are a potential correlate of protection in their study, but this is in fact not supported by the data, and any associations of T cells and protection only apply in a particular non-representative subset analysis:

That is, the authors conclude that “the magnitude of vaccine-induced neutralizing activity and specific CD4 T-cells after bivalent vaccination may serve as a correlate for protection in previously non-infected individuals. (Abstract, line 37)” The unwary reader may understand this to mean that there is evidence T cells may be correlated with protection (generally), but

this is in fact not what the authors are saying. The authors observation regarding a potential correlation, only apply to one subset of the overall population, and does not highlight that more generally T cells were not correlated with protection.

This is a critical point, because across the whole population the conclusion seems completely opposite.

i.e.: the authors observe lower risk of infection in previously infected than in naïve individuals. However, they see no difference in T cell responses between these groups. So this would directly tell us that T cells are not different when protection is clearly different. In this case, neutralisation titer does seem to predict the difference in protection (as expected and is established in the field).

- The fact that the authors find T cells are a correlate in people with breakthrough infection ONLY in the naïve group means that even if this were correct, this is not a useful correlate. E.g. If you need to know the infection history of an individual as well as the immunological measurements, then these are not very useable correlates of protection (particularly since now essentially everyone has been previously infected). Thus, it is inappropriate to look for correlates in every subset of individuals, instead a useful correlate is one that will work across all your subgroups.

- Since antibodies are already an established correlate, an additional correlate such as T cells responses would need to add predictive value over the existing correlate. I.e: because neutralization titers and T cell responses are correlated, T cells will look correlated but not be adding any information other than as (poor) predictors of the neutralisation titer. The authors do not test this. i.e. They must analyse the entire cohort (not subgroups) for whether neutralising antibodies predict breakthrough infection and then see whether adding a T cell measure improves this prediction significantly (in a nested model). They also must adjust for multiple comparisons and control for the number of predictors they test for. The authors will quickly find that their study is not powered to look for T cell correlates – despite this being the motivation of the study. This is particularly made difficult because neutralising antibodies correlate with T cell responses – and untangling these processes is a major challenge for the field that this study makes no attempt to resolve.

Also, as has been emphasised by many including the authors, T cells are, if at all, likely a correlate of severe disease and not of protection from infection (but the authors can only consider the latter here).

2) Additionally, the study is lacking sufficient novelty because there are numerous other similar studies. The authors argue their study is distinct in three main ways:

a) The authors suggest few studies look at T cell immunogenicity after bivalent vaccines, or only have small sample sizes.

But there are many other studies that report on T cell responses after bivalent vaccination – some with bigger sample sizes. And these are not mentioned or considered by the authors, eg.

<https://onlinelibrary.wiley.com/doi/full/10.1002/jmv.28989>

<https://www.sciencedirect.com/science/article/pii/S1473309923001408>

[https://www.thelancet.com/journals/lanmic/article/PIIS2666-5247\(23\)00105-2/fulltext](https://www.thelancet.com/journals/lanmic/article/PIIS2666-5247(23)00105-2/fulltext)

<https://www.sciencedirect.com/science/article/pii/S0163445323002037>

b) The authors suggest not many studies have looked for T cell correlates.

But there are more appropriately designed studies that have looked at this (e.g. Kemlin et al., American Journal of Transplantation 2023) – some of which have found an association, but all suffering from the problem that antibodies are correlated with T cells and so it is very difficult to disentangle these.

c) The authors suggest they find a unique observation of CD4 T cells being a potential correlate of protection.

But this is not generally applicable conclusion, and likely arise from confounding due to the co-correlation of T cells with neutralising antibodies (see above).

Minor points:

1) The authors argue in the introduction that “SARS-CoV-2 specific T-cells, which have been shown to mediate protection from severe disease^{12,13}” – and they cite two review articles – one of which argues the opposite - that there is no such evidence...

2) Line 107, 116, etc says “3 log-levels” – this is not the correct way to describe the difference since it is actually 3 log₂ level (i.e. 3 lots of 2 fold). They authors should just say 8 fold.

3) Figure 2 and 5: IC₅₀ is a titer and not a percentage - so remove “[%]” from y axis label.

It is this reviewers opinion that the major conclusions of the paper are, at best, incremental advances and at worst misleading for the unwary reader. For CD4 T cells to be established as a correlate of protection they would require a lot more than the an uncorrected correlation (not adjusting for co-correlation of nAb and T cells) in a subset of individuals demonstrated by the authors here. The authors should remove all statements that their work may indicate CD4 T cells as a correlate of protection. In fact, some results of their work actually indicate CD4 T cells are not a useful correlate. i.e. T cells responses are the same in the two groups that ultimately have different risks (previously infected and non-infected), but neutralization titers are different between the groups.

Responses to REVIEWER COMMENTS

Reviewer #1

Urschel et al report on antibody titres and CD4/CD8 T cell responses following vaccination with BA4/5 bivalent mRNA booster vaccines. Among the cohort, the group with prior COVID-19 exhibited higher antibody titres both before and after vaccination compared with uninfected participants. Spike-specific T cell frequencies were boosted by vaccination but were relatively similar regardless of infection history. Over the course of 5 months of follow-up, breakthrough infections were observed among both groups. There was no identified correlate of protection among individuals with prior COVID, but the uninfected group exhibited low post-vaccination neutralising antibody titres and CD4 T cell responses in individuals with subsequent breakthrough infection. Overall, these observations are consistent with other reports in the literature, reflecting the immunogenicity of the bivalent booster vaccines and the ability of S-specific T cells to recognise both ancestral and omicron-related spike proteins.

We thank the reviewer for the appreciation of our work and for this succinct summary of our data. Although some parts of our results have already been described in other studies (in part with previous generations of vaccines), we feel that our manuscript adds novelty as we describe a head-to-head analysis of reactogenicity and immunogenicity of the BA.4/5-bivalent vaccine in individuals with and without prior infection, and address both humoral and cellular immunity (including CD4 and CD8 T cells) in the same study. Of particular interest are differences in the correlations between CD4 and CD8 T cells, and neutralizing antibodies in individuals with and without prior infection, which may also have implications for subsequent breakthrough infections.

Can the authors please provide further explanation and references to justify the use of CTLA-4 as a marker of recent antigen exposure (line 157)?

CTLA-4 is known to be upregulated after antigen encounter during infections or vaccinations. An example is a previous study in patients with herpes zoster, where VZV-specific T cells show an increased expression of CTLA-4, which decreases after resolution of symptoms (Schub et al. 2015; Schub et al. 2018). Vaccine-induced upregulation of CTLA-4 expression was also shown on influenza-specific T cells during the first two weeks after influenza vaccination, and decrease over the following weeks (Ledo et al. 2019). Other examples include observations of specific immunity in patients with chronic HIV or active CMV infection (Kaufmann et al. 2007; Dirks et al. 2013), where persistently high levels of antigen-specific T cells with this phenotype are associated with functional energy and sustained viral replication. We have now given references as suggested in the results section (p. 9), and expanded the discussion on CTLA-4 on p. 13 accordingly.

As the antigen-specific cells have been re-stimulated in vitro, can it be shown that CTLA-4 expression has not been induced by this antigen exposure? How long does CTLA-4 expression persist after antigen exposure (i.e. what timeframe does 'recent' reflect)?

It is unlikely that the high CTLA-4 expression is a result of upregulation in vitro, as CTLA-4 expression on spike-specific CD4 and CD8 T cells in samples prior to vaccination was significantly lower despite similar stimulation procedure. Moreover, no increase was observed for CD4 or CD8 T cells after

stimulation with SEB. Together this indicates that the upregulation of CTLA-4 is specific for SARS-CoV-2 reactive T cells and is associated with the vaccination in vivo. We have added another figure (now supplementary **figure S4**) to emphasize this more clearly. We are unable to provide information, as to how long CTLA-4 expression persists after antigen-exposure, because we have not included further time points in the present study. However, based on previous studies mentioned above, CTLA-4 expression on vaccine-induced influenza-specific T cells reverted back to pre-vaccine levels after 6 months, and CTLA-4 expression on VZV-specific T cells after acute herpes zoster was decreasing after 3 months. Discussion on the dynamics of CTLA-4 expression on antigen-specific T cells are now included on **p. 13** (see previous comment).

The longitudinal tracking of breakthrough infection following vaccination is interesting and provides a good opportunity to prospectively assess potential protective immune responses. While the observation that individuals experiencing infection tended to have lower neutralising antibody responses after vaccination, it would be useful to try to understand the factors that underpin these outcomes. Did the breakthrough individuals have particularly poor fold-change responses to vaccination? Did they have low pre-vaccine titres? Was the post-vaccination breadth of those responses less compared to the non-breakthrough cohort? Similarly, did they exhibit differences in their pre-vaccine CD4 T cell responses, or have relatively poor cTFH responses?

We thank the reviewer for this comment towards analyzing other factors that may be associated with breakthrough infections after vaccination. As suggested by the reviewer, we have looked at both baseline IgG-levels and neutralizing antibody levels stratified for individuals with or without breakthrough infection. As shown in supplementary **figure S5a**, we did not find any differences in baseline IgG-levels or in the fold increase of IgG levels. Interestingly, however, baseline neutralizing antibody levels in individuals with subsequent breakthrough infections were numerically lower with significant differences for neutralizing antibodies towards the parental strain, and some omicron variants of concern (**figure S5b**): In previously non-infected individuals with subsequent breakthrough infections, antibody-mediated neutralization levels towards parental ($p=0.008$) and BA.2 ($p=0.029$) strain were lower; likewise, previously infected individuals with subsequent breakthrough infections had lower neutralization titers towards the parental ($p=0.014$), BA.1 ($p=0.020$) and BA.4/5 ($p=0.009$) strains. This finding further underlines the role of neutralizing antibodies in protective immunity, although the associations with breakthrough infections were more pronounced for post-vaccination titers (shown in **figure 6** of the main manuscript).

We also analyzed baseline CD4 and CD8 T-cell levels in individuals with and without subsequent breakthrough infection (supplementary **figure S5c**). In line with post-vaccination results, non-infected individuals with subsequent breakthrough infection had lower levels of spike-specific CD4 T cells than individuals without subsequent infection ($p=0.041$). In contrast, among previously infected, baseline CD4 T-cell levels did not differ among those who did or did not experience a further infection. Baseline CD8 T-cell levels generally showed a higher variability. The only difference we observed was that previously infected individuals with subsequent breakthrough infection had significantly higher levels of spike-specific CD8 T cells than individuals without breakthrough infection ($p=0.016$). We did not

observe any impact of the fold-change of CD4 or CD8 T-cell levels on subsequent breakthrough infection.

We have now added these data on pre-vaccine IgG, neutralizing antibodies as well as fold changes in a new supplementary **figure S5** and referred to this figure in the results section (**p. 10**).

Regarding post-vaccination breadth, our results in figure 6 show lower levels of neutralizing antibodies and CD4 T cells in previously non-infected individuals, but this equally applied to the parental strain and all tested omicron variants of concern.

It would be interesting to relate our findings to spike-specific follicular helper T cells (based on CXCR5 expression and/or expression of IL-21), which we unfortunately did not determine in the present study due to limited sample volume and in light the low abundance of follicular helper T cells in peripheral blood (i.e. Ledo *et al.* 2019). We have acknowledged the lack of Tfh cell analysis in the limitations section on **p. 16**.

Reviewer #2

The manuscript by Urschel et al. seeks to determine the impact of bivalent mRNA vaccines in individuals previously infected or not with SARS-CoV-2. The authors found a higher vaccine-induced IgG response in individuals with prior infection compared to non-infected. In contrast, a lower difference was shown in CD4 and CD8 responses. Breakthrough infections were more frequent in individuals with no previous infection who had low humoral and cellular immunity. The data looks interesting and bring the importance of vaccine-induced cellular immunity on subsequent infection and as a possible correlate of protection. There are minor points to be addressed.

We thank the reviewer for the succinct summary and for the appreciation of the relevance of our work.

Line 80 – It would be interesting to add the interval between the infection to the bivalent vaccination (in the “infected group”) as it can influence the rapid response 2 weeks after vaccination.

We have followed the suggestion of the reviewer. We had already mentioned the infecting strain in **table 1** (with most infections having occurred in the omicron era). We have now added the median time from infection to vaccination for the infected group, which was 206 (IQR 49) days (from 60 to 869 days, **table 1, p. 29**). We did not find any significant correlation of the time since infection with any of the immunological parameters, which may in part be due to the fact that the different infecting strains may by itself have a variable impact on spike-specific immunity (briefly included in the discussion on **p. 14**).

Line 141 - Have the authors tried to correlate CD8 and CD4 responses out of activated cells (i.e., no CD69+)? Even though gating on activated cells string the analysis, I'm questioning if you could miss some specific cytokines as the frequency of both CD4 and CD8 is low.

Please note that we hardly observe any cytokine-producing cells that are CD69 negative. This is illustrated in the representative dot plot of the individual shown in **figure S2** (demonstrating IFN γ producing cells) and in **figure S7** showing IL-2 and TNF α producing cells. Thus, it is considered

unlikely that restriction to CD69 positive T cells would substantially lead to an underestimation of spike-specific T-cell levels. To illustrate this further, we have re-analyzed 50 stimulatory reactions from 10 individuals with and without consideration of CD69 positivity. Among all antigen-specific IFN γ positive events, only a median of 0.11% (IQR 1.64%) of CD4 T cells and 0% (IQR 1.03%) of CD8 T cells were CD69 negative, which further substantiate that it is unlikely that we miss any specific cells. We have added the rationale to include CD69 in the methods section (p. 19).

Line 152 – The use of CD40L instead of CD69 would better represent the correlation between IgG and CD4s.

Regarding correlations between specific CD4 T cells and IgG, we would like to emphasize that we do see a correlation between the two parameters, but this was only observed in previously non-infected individuals (see figure 4). Thus, this indicates that the lack of correlation is unlikely based on technical reasons but rather influenced by prior infection.

Nevertheless, we have explored technical issues further: In our study, we chose to use CD69 and intracellular cytokine analysis as one of the state-of-the-art methods for the simultaneous detection of antigen-specific CD4 and CD8 T cells from the same stimulatory reaction. Unfortunately, we do not have any stored samples to perform CD40L staining from the same data set. However, to compare both readouts, we performed comparative analyses of CMV-specific CD4 T cell levels after stimulation of whole blood samples from 12 CMV-seropositive individuals and did not find any significant differences, as shown in the following figure:

Therefore, we feel it is rather unlikely that using CD40L as an alternative staining procedure would have yielded different results.

Line 165 – Have the authors tried to analyze the percentage of individuals who are producing 1, 2, or 3 cytokines? This data could be interesting to improve the functional potency of the T cells. mRNA induces a Th1 profile preferentially, but considering the heterogeneous of the individuals, it also is interesting to analyze Th2 cytokines

There may have been a misunderstanding, but it should be noted that analysis of the *percentage of individuals* who are producing 1, 2, or 3 cytokines is not possible, as each individual has spike-specific T cells expressing either 1, 2, or 3 cytokines alone or in combination. This results in a total of 7 subpopulations which are currently shown in **figure 5**. We would like to keep this type of analysis, as displaying subpopulations producing 1, 2 or 3 cytokines would lead to some loss of information on the response patterns of the cells with their individual cytokines. However, to follow the reviewer's suggestion, we have now changed the order of the 7 subpopulations in figure 5 and labelled the respective subpopulations producing 3, 2 and 1 cytokine.

For reasons pointed out by the reviewer, we have restricted our analysis to Th1 cells, without being able to assess Th2 cytokines from the same dataset. We have now included this as a limitation in the discussion on **page 16**.

Line 345 – How was determine the cut-off of CD4 and CD8 (0.03%)? Was it the same for both?

We thank the reviewer for this comment. In general, the cut-off of 0.03% was established in previous studies based on the distribution of CD4 T-cell frequencies after control stimulations, and was also displayed as a cut-off for CD8 T-cell populations. In response to the reviewer's comment we have now similarly determined the cut-off from for CD8 T-cells. This revealed a cut-off of 0.06% which we have now displayed in the figures. Please note, that this change in the cut-off for CD8 T cells does not have any implications for our results, as we only compared quantitative levels of reactive T cells in the various groups. A description of how the cut-offs were determined is now included in the methods section (**p. 19**).

Line 38-39 – The wording correlation of protection in the aspect of the magnitude of CD4s is "risky". It would be better to use "an indicator".

We thank the reviewer for this suggestion, which we have followed. We have also revised the concluding sentence in the abstract (**p. 3**), and systematically checked the wording in that regard to reach a balanced description and discussion of our data (also in response to reviewer 3).

Figure 5 – It would be great to have the N of each group indicated in the graphs (for example: right below the dots). The visualization of sample size would be more informative in statistics analysis.

We have now added the sample size in each of the panels and/or the figure legends.

On extended Figure 3, IgG was correlated with CD4 in infected but was not in uninfected. Have the authors tried to perform this analysis with the "no BTI" group to see if the CD4 and IgG had a coordinated response towards protection?

-Extended figure 3 is now figure 4-

Please note that IgG and CD4 T-cell levels were correlated in the **uninfected** (and not in the infected, i.e. vice versa). We have now looked at the correlation between IgG and CD4 T-cell levels in the "no BTI" group (individuals who did not experience any breakthrough infection irrespective of prior infection history). We indeed see a correlation between both parameters for CD4 T cells and IgG, CD4

T cells and neutralizing antibodies towards the parental strain and for BA.4/5 specific CD4 T cells and neutralizing antibodies toward BA.5. This correlation matrix is now included as **figure S6g**.

The percentage of BTI was more frequent in the non-infected group and in those who had low humoral and cellular immunity two weeks after bivalent mRNA vaccination. The authors have mentioned that the magnitude of vaccine-induced neutralizing activity and specific CD4 T cells may serve as a correlate for protection. However, no further analysis with the indicators was performed – how are the magnitude reference value for protection (CD4 frequency, IgG titer/activity, etc)? Additionally, the sample size for “BTI yes” seems too low to determine that. I would use “an indicator of protection” instead.

In general, we would prefer a descriptive analysis of quantitative differences in neutralizing antibody titers and CD4 T-cell levels, also in light of comments towards a more cautious wording around “correlates of protection” in response to reviewer 3 and to the editor. We have therefore also used “indicator of protection” as terminology as suggested by the reviewer. We agree with the reviewer that the sample size of BTI is rather low to provide meaningful cut-offs as indicator for protection. However, we added AUC values derived from regression analyses also in response to reviewer 3 (now supplementary **table S3**).

Reviewer #3

The authors present a well written study of a cohort of individuals (n=127) who received an Omicron BA.4/5 bivalent booster vaccine. The authors present assessments of immunogenicity and compare these between those who had previous infection (before the boost) and those who did not report having prior infection (and had N-negative serology). Unsurprisingly they find that those with previous infection had higher neutralising antibody titers (but a smaller boost in titer) – consistent with other reports. They then examine breakthrough infections in these cohorts and report associations between neutralizing antibodies and CD4 T cells and risk of breakthrough infections (but only in the subset of those not previously infected). From this the authors draw somewhat problematic conclusions about “correlates of protection”.

Unfortunately, given the number of other very similar studies, and problematic conclusions around correlates, this study presents a fairly incremental advance on the existing work in the field.

General remarks to reviewer 3’s comments:

Overall, we see a particular strength of our study in providing data on the differences between individuals with and without previous infections. While our main focus was on reactogenicity and immunogenicity of the BA.4/5 bivalent vaccination in the two groups, studies that have clearly distinguished between individuals with and without history of infection when looking at the association of vaccine-induced antibodies and T cells with subsequent breakthrough infection are still limited. As we realized that many comments by reviewer 3 were centered around the question of which findings that we describe would also apply to the whole group of 127 individuals (without stratification into subgroups of individuals with and without prior infection), we have prepared additional display items, where the immune parameters are shown for the whole group, and have

added regression analyses (odds ratios and AUC). We feel that these additional analyses will also better emphasize that some effects are only unmasked after stratification (**see point-by-point comments below**).

We feel that the comparative and separate look at these two distinct groups is relevant in light of the fact that more and more individuals will soon have a history of infection, whereas most initial vaccine studies, including the first data on neutralizing antibodies as potential correlates of protection, were primarily restricted to non-infected individuals. Thus, it is reassuring that our results of previously non-infected individuals are in line with many publications on vaccine-induced immune responses of the last years predominantly generated among infection-naïve individuals. In contrast, results of our group of individuals with a history of prior infection are distinct and in part also unexpected, especially as CD4 T cells and neutralizing antibodies did not correlate, and as neither levels of CD4 T-cells nor of neutralizing antibody levels differed between individuals with and without breakthrough infection. We believe that our data will provide an interesting basis for future studies on the role of vaccine-induced antibodies and T cells for subsequent infection in individuals with a history of infection, who is likely the dominant population by now.

We would also like to emphasize that our observational study populations are relatively unique in that they were recruited at a time where knowledge of infection history was still very reliable, as antigen- and/or PCR-testing was still common practice. Nowadays, this type of comparative study would be difficult, as the percentage of clearly defined non-infected individuals is substantially decreasing, and the lack of prior infection is difficult to prove after abandoning mandates for regular SARS-CoV-2 testing.

Reply to specific comments

Comment 1:

Specifically, this reviewer's greatest concern is that the unwary reader may understand the authors as having shown that T cells are a potential correlate of protection in their study, but this is in fact not supported by the data, and any associations of T cells and protection only apply in a particular non-representative subset analysis: That is, the authors conclude that "the magnitude of vaccine-induced neutralizing activity and specific CD4 T-cells after bivalent vaccination may serve as a correlate for protection in previously non-infected individuals. (Abstract, line 37)" The unwary reader may understand this to mean that there is evidence T cells may be correlated with protection (generally), but this is in fact not what the authors are saying. The authors observation regarding a potential correlation, only apply to one subset of the overall population, and does not highlight that more generally T cells were not correlated with protection. This is a critical point, because across the whole population the conclusion seems completely opposite. i.e.: the authors observe lower risk of infection in previously infected than in naïve individuals. However, they see no difference in T cell responses between these groups. So this would directly tell us that T cells are not different when protection is clearly different. In this case, neutralisation titer does seem to predict the difference in protection (as expected and is established in the field).

The reviewer states that our description of results and phrasing is correct, which is reassuring, but the reviewer raises some concern that readers may misunderstand our results from the previously non-infected group as a general finding applicable for both previously infected and non-infected individuals. This is indeed not the case and was clarified throughout the manuscript. We have also revised terminology and avoided the term 'correlate of protection' as suggested (in the concluding sentence in the abstract and throughout the manuscript).

As suggested by the reviewer, we have prepared further analyses and display items on the whole population of 127 individuals (without stratification into previously infected and non-infected), which are now included as supplementary **figures S3** (and figure S6, see comment below). As established in the field, neutralizing titers seem to have predictive value for protection. The reviewer is correct in stating that there are no differences in T-cell levels between previously infected and non-infected individuals (seen in figure 3). However, despite identical quantitative levels of T cells in both groups, the correlation matrices indicate that previously infected and non-infected individuals show clear differences regarding correlations between CD4 and CD8 T cells (seen in figure 3d), between CD4 T cells and neutralizing antibodies, and between CD8 T cells and neutralizing antibodies (seen in figure 4). We therefore feel that this stratification is important to clearly appreciate differences between individuals with and without prior infection, and we hope that these additional analyses will better emphasize that some effects are only unmasked after stratification.

To follow the reviewer's recommendation, we also have included regression analyses including odds ratios and AUCs (supplementary **tables S2** and **S3**). We also acknowledge that the number of individuals with breakthrough infections is low, and we have revised the wording regarding correlates of protection to avoid misunderstanding. Our data from the two subgroups of individuals with and without breakthrough infection should be considered as a basis for future studies with larger sample sizes. Given the ongoing infection waves with SARS-CoV-2, where most individuals will soon have a history of infection, studies specifically looking at individuals with prior infection can readily be carried out to corroborate our in part unexpected findings of the infected subgroup. Data are described in the results section (**p. 10/11**) and a revised discussion including limitations is found on **p. 14-17**.

- The fact that the authors find T cells are a correlate in people with breakthrough infection ONLY in the naïve group means that even if this were correct, this is not a useful correlate. E.g. If you need to know the infection history of an individual as well as the immunological measurements, then these are not very useable correlates of protection (particularly since now essentially everyone has been previously infected). Thus, it is inappropriate to look for correlates in every subset of individuals, instead a useful correlate is one that will work across all your subgroups.

Our primary focus was the characterization of differences in vaccine-induced immune responses in two well defined groups of individuals with and without prior infection, and we provide an objective description of our results. We do not see any reason to assume that our results are not correct, and the question whether or not our results are useful should be addressed in the discussion, which we have now done in more detail. Regarding "usefulness", we agree with the reviewer that now essentially everyone has been previously infected. We are convinced that this gives a particular relevance to the

separate analysis of the two groups to specifically characterize immunological differences between individuals without prior infection and individuals with infections that nowadays should be the dominant population (see discussion **p. 14-17**).

- Since antibodies are already an established correlate, an additional correlate such as T cells responses would need to add predictive value over the existing correlate. I.e. because neutralization titers and T cell responses are correlated, T cells will look correlated but not be adding any information other than as (poor) predictors of the neutralisation titer. The authors do not test this. i.e. They must analyse the entire cohort (not subgroups) for whether neutralising antibodies predict breakthrough infection and then see whether adding a T cell measure improves this prediction significantly (in a nested model). They also must adjust for multiple comparisons and control for the number of predictors they test for. The authors will quickly find that their study is not powered to look for T cell correlates – despite this being the motivation of the study. This is particularly made difficult because neutralising antibodies correlate with T cell responses – and untangling these processes is a major challenge for the field that this study makes no attempt to resolve.

We agree with the reviewer that there is merit to validating this finding in a larger sample in diverse settings. We have further noted the need for further validation research in the discussion section. In addition, as stated above, we have followed the suggestion of the reviewer and performed multivariate logistic regression analysis in all individuals (n=127), and in the two groups of individuals without (n=63) and with previous infection (n=64) to assess the relationship between immunological parameters (spike-specific CD4 T cells and neutralizing antibodies) and occurrence of breakthrough infections. These data confirm published evidence and the reviewer's notion that neutralizing antibodies were the best predictors of breakthrough infection when looking at the whole cohort. However, multivariate logistic regression analyses indicate differences when separately looking at individuals with and without prior infection (**tables S2 and S3**). We have included the analyses in the results section (**p. 10/11**) and have discussed them, including limitations and need for future studies (**p. 14-17**).

We would like to point out that we used the same analysis approach as in the study by Kemlin *et al.* in transplant recipients after the third monovalent vaccination, which the reviewer pointed out as "more appropriately designed to look at this" (**see comment 2b below**). Both the Kemlin study and our analysis made sure to avoid multicollinearity, hence should be able to reveal independent effects. The sample size of the Kemlin study was lower than each of our two groups (n=53), and individuals with prior infection were explicitly excluded. Data are very much in line with our subgroup results of previously non-infected individuals. Together with the contrasting finding that neither antibodies nor T cells differed in previously infected individuals with and without subsequent breakthrough infections, we feel that the separate analysis is relevant to be reported.

Also, as has been emphasised by many including the authors, T cells are, if at all, likely a correlate of severe disease and not of protection from infection (but the authors can only consider the latter here).

As we do not have any individuals with severe disease in our cohorts, this aspect cannot be addressed further.

Comment 2:

Additionally, the study is lacking sufficient novelty because there are numerous other similar studies. The authors argue their study is distinct in three main ways:

a) The authors suggest few studies look at T cell immunogenicity after bivalent vaccines, or only have small sample sizes. But there are many other studies that report on T cell responses after bivalent vaccination – some with bigger sample sizes. And these are not mentioned or considered by the authors, eg...

Please note that our manuscript was submitted to Nature communications on the 12th of June 2023. We have carefully reviewed the literature to acknowledge work which has been done up to this time point. We were still not aware of any other study of similar or bigger sample size that has looked at reactogenicity and immunogenicity before and after BA.4/5 bivalent vaccination including IgG, neutralizing antibodies, CD4 and CD8 T cells in individuals with and without prior infection. Overall, the four papers mentioned by the reviewer were either published after submission, used vaccines other than the bivalent BA.4/5 vaccine or only tested pre-vaccination samples, or only had limited immune parameters tested from far lower numbers of individuals. With all due respect for the value of each individual study cited, their scope is different from our study.

We comment on each of the four paper as follows:

(1) Mak, W.A., *et al.* Ancestral SARS-CoV-2 and Omicron BA.5-specific neutralizing antibody and T-cell responses after Omicron bivalent booster vaccination in previously infected and infection-naïve individuals. *J Med Virol* **95**, e28989 (2023).

<https://onlinelibrary.wiley.com/doi/full/10.1002/jmv.28989>

Please note that this manuscript was first published on the 10th of August, i.e. appeared 59 days after our submission. It is interesting to be considered, but should not be used as a claim for limited novelty (which is also in line with Nature Communications editorial policy published in 2020 (Strength in numbers | Nature Communications), where the following statement can be found in the *entrée*: "...At Nature Communications, we commit to disregard from our editorial evaluation any competing works that are published while a submission to our journal is under review or under revision by the authors...").

While the paper is interesting, we would like to note that the paper analysed the BA.1 bivalent vaccine. Moreover, sample size was smaller (59 individuals including infected and non-infected individuals, of which only subgroups of 18 or 25 receiving the vaccine were further analysed). T-cell analyses were performed using ELISPOT assay and were restricted to one Omicron strain. Therefore, no information on CD4 and CD8 T cells and reactivity towards other strains is available from this study. Moreover, sampling was 57 days (IQR 38-65) after vaccination, and two different vaccine types (Moderna and Biontech/Pfizer) were used, which by itself may cause variability. We included this study in the discussion on **p. 12**.

- (2) Tan, N.H., *et al.* Immunogenicity of bivalent omicron (BA.1) booster vaccination after different priming regimens in health-care workers in the Netherlands (SWITCH ON): results from the direct boost group of an open-label, multicentre, randomised controlled trial. *Lancet Infect Dis* **23**, 901-913 (2023). <https://www.sciencedirect.com/science/article/pii/S1473309923001408>

This manuscript was published in August (online on the 20th April) and analysed reactivity and immunogenicity of the BA.1 bivalent vaccine in 187 individuals. The focus of this study was on comparative analysis of two mRNA vaccines (Biontech/Pfizer vs. Moderna) depending on the priming regimen. Individuals with infections were explicitly excluded. Although this study is interesting and scientifically sound, we do not see any overlap with our research question.

- (3) Traut, C.C. & Blankson, J.N. Bivalent mRNA vaccine-elicited SARS-CoV-2 specific T cells recognise the omicron XBB sublineage. *Lancet Microbe* **4**, e388 (2023). [https://www.thelancet.com/journals/lanmic/article/PIIS2666-5247\(23\)00105-2/fulltext](https://www.thelancet.com/journals/lanmic/article/PIIS2666-5247(23)00105-2/fulltext)

This is a half-page letter with one figure describing BA.4/5 bivalent vaccine-induced T-cell reactivity towards the parental strain and the Omicron XBB strain from 21 individuals. This study explicitly excluded individuals who had a known infection within the last three months (but unknown whether individuals have a history of infection more than three months ago). T-cells were analysed using an ELISPOT assay, hence no distinction was made between CD4 and CD8 T cells. Moreover, antibody data were not included in parallel. Again, this study is considered timely and well done, but the sample size is not higher. Due to the short format, no information is given regarding sampling time after vaccination.

- (4) Pighi, L., Henry, B.M., De Nitto, S., Salvagno, G.L. & Lippi, G. Cellular immunity against SARS-CoV-2 depends on the serological status. *J Infect* **87**, 57-58 (2023). <https://www.sciencedirect.com/science/article/pii/S0163445323002037>

This one page letter analyses cellular immunity with an IGRA assay (secretion of IFN γ in the supernatant) in 78 individuals with (n=35) and without (n=39) a positive NCAP serology indicative of prior infection. Please note that this analysis was performed BEFORE bivalent vaccination and includes just one figure on IGRA responses in the two groups. No post-vaccination data are shown. The scope of this study is clearly different from ours.

b) The authors suggest not many studies have looked for T cell correlates. But there are more appropriately designed studies that have looked at this (e.g. Kemlin et al., American Journal of Transplantation 2023) – some of which have found an association, but all suffering from the problem that antibodies are correlated with T cells and so it is very difficult to disentangle these.

We thank the reviewer for highlighting the work by Kemlin and coworkers performed *in renal transplant recipients*, which was performed after a third dose of the *monovalent vaccine*. As outlined above, this study found a correlation between antibodies and T cells, and both parameters showed independent predictive value for breakthrough infection in multivariate analyses. An important detail is the fact that this study has only included individuals without prior infection. As with many studies published so far, individuals with a history of SARS-CoV-2 infection were explicitly excluded. As stated

above, findings of the study by Kemlin *et al.* correspond to what we found after bivalent vaccination in the group of previously non-infected individuals. Both the Kemlin study and our analyses shown in tables S2 and S3 in response to the reviewers paid specific attention to avoid multicollinearity in the analyses.

Neither the study by Kemlin *et al.* nor any other study with larger sample size performed a parallel analysis of both antibodies and T cells among previously infected individuals in direct comparison with non-infected.

We have now included Kemlin *et al.* in our discussion to illustrate similarity with our results among previously non-infected, and emphasized the parallel recruitment of individuals with history of infection as a particular asset of our study, especially as results are different from previously non-infected. The study is now included in the discussion (p. 15).

c) The authors suggest they find a unique observation of CD4 T cells being a potential correlate of protection. But this is not generally applicable conclusion, and likely arise from confounding due to the co-correlation of T cells with neutralising antibodies (see above).

We believe that the revisions in response to the previous comments have addressed these comments, namely by avoiding the term “correlates of protection” (also in the abstract), by regression analyses that also addressed concerns on collinearity, and by emphasizing the need for future studies.

Minor points:

1) The authors argue in the introduction that “SARS-CoV-2 specific T-cells, which have been shown to mediate protection from severe disease^{12,13” – and they cite two review articles – one of which argues the opposite - that there is no such evidence...}

We apologize and we have corrected the sentence accordingly. We had cited two review articles of which one is illustrating evidence for T cells as mediator of protection from severe disease, and the other one nicely illustrates the theoretical rationale and the difficulties in dissecting the exact contribution of T cells for protection from infection and disease including avenues towards how future research could provide more information on the specific role of T cells. We have now modified the sentence to read “which are discussed to have a potential role in protection from severe disease” (p. 4).

2) Line 107, 116, etc says “3 log-levels” – this is not the correct way to describe the difference since it is actually 3 log₂ level (i.e. 3 lots of 2 fold). They authors should just say 8 fold.

We thank the reviewer for this comment and have changed this throughout the manuscript.

3) Figure 2 and 5: IC50 is a titer and not a percentage - so remove “[%]” from y axis label.

We thank the reviewer for this comment and have changed this accordingly.

It is this reviewers opinion that the major conclusions of the paper are, at best, incremental advances and at worst misleading for the unwary reader. For CD4 T cells to be established as a correlate of protection they would require a lot more than the an uncorrected correlation (not adjusting for co-correlation of nAb and T cells) in a subset of individuals demonstrated by the authors here. The authors should remove all statements that their work may indicate CD4 T cells

as a correlate of protection. In fact, some results of their work actually indicate CD4 T cells are not a useful correlate. i.e. T cells responses are the same in the two groups that ultimately have different risks (previously infected and non-infected), but neutralization titers are different between the groups.

See above, we hope that the revisions in response to the previous comments have addressed these comments, namely by avoiding the term “correlates of protection” (also in the abstract), by regression analyses that also addressed concerns on collinearity, and by emphasizing the need for future studies.

REVIEWER COMMENTS

Reviewer #3 (Remarks to the Author):

I thank the authors for toning down the references to correlates within this manuscript. The authors are careful and accurate in their reporting of what they have done. Overall, this reviewer's main concern remains that it is unclear what the primary innovations of this work are – a part from reporting a panel of immunological measurements in a particular cohort? It seems very niche (i.e. to do with correlates of breakthrough infection in BA.4/5 bivalent vaccinated and previously uninfected individuals?). But judgement as to the sufficient novelty of this work is best left with the editor. Regarding statistical correctness of the work, I have one remaining major concern. That is, the primary conclusion of the work is not supported by the statistical analysis (see below).

Statistical concern:

In the revision the authors have shifted the focus of the manuscript, but the conclusion is not directly supported by the current analysis.

The new primary conclusion highlighted in the abstract (and argued as the strength of the study in the response to reviewers) is now:

“Thus, immune responses after BA.4/5-adapted bivalent vaccination DIFFER BETWEEN INDIVIDUALS WITH AND WITHOUT PRIOR INFECTIONS, which may have implications for occurrence of subsequent breakthrough infections.” [Reviewer's emphasis]

i.e. the authors now emphasise the primary novelty of their study is that they find a “difference” in the immunological associations between the previously infected group and previously uninfected groups. By this the authors appear to mean that they see a correlation with neutrophils and T cells and breakthrough infection (Table S2) in previously uninfected group, which they do not see in the previously infected group. But the authors do not actually test whether there is evidence of a statistical difference between these groups. What I mean is

that in statistics, if you want to show there is evidence of a difference between two groups you should report a statistical test with a null hypothesis that the groups are the same. The authors have not done this, they have only analysed the two groups separately. To test if there is a difference in association between the groups the authors should extend their multiple regression of the whole cohort and add prior infection status as an interaction term on the predictor to determine if there is evidence of a difference between the groups (e.g. whether there is a significant interaction between infection status and T cell level, etc).

As it stands the authors have not tested whether there is actually evidence of different immunological associations with protection between the previously infected and uninfected groups.

The authors must remove this conclusion - or add the appropriate statistical tests.

Comment of reviewer #3

I thank the authors for toning down the references to correlates within this manuscript. The authors are careful and accurate in their reporting of what they have done. Overall, this reviewer's main concern remains that it is unclear what the primary innovations of this work are – a part from reporting a panel of immunological measurements in a particular cohort? It seems very niche (i.e. to do with correlates of breakthrough infection in BA.4/5 bivalent vaccinated and previously uninfected individuals?). But judgement as to the sufficient novelty of this work is best left with the editor. Regarding statistical correctness of the work, I have one remaining major concern. That is, the primary conclusion of the work is not supported by the statistical analysis (see below).

Statistical concern: In the revision the authors have shifted the focus of the manuscript, but the conclusion is not directly supported by the current analysis.

The new primary conclusion highlighted in the abstract (and argued as the strength of the study in the response to reviewers) is now:

“Thus, immune responses after BA.4/5-adapted bivalent vaccination DIFFER BETWEEN INDIVIDUALS WITH AND WITHOUT PRIOR INFECTIONS, which may have implications for occurrence of subsequent breakthrough infections.” [Reviewer's emphasis]

i.e. the authors now emphasise the primary novelty of their study is that they find a “difference” in the immunological associations between the previously infected group and previously uninfected groups. By this the authors appear to mean that they see a correlation with neutrophils and T cells and breakthrough infection (Table S2) in previously uninfected group, which they do not see in the previously infected group. But the authors do not actually test whether there is evidence of a statistical difference between these groups. What I mean is that in statistics, if you want to show there is evidence of a difference between two groups you should report a statistical test with a null hypothesis that the groups are the same. The authors have not done this, they have only analysed the two groups separately. To test if there is a difference in association between the groups the authors should extend their multiple regression of the whole cohort and add prior infection status as an interaction term on the predictor to determine if there is evidence of a difference between the groups (e.g. whether there is a significant interaction between infection status and T cell level, etc). As it stands the authors have not tested whether there is actually evidence of different immunological associations with protection between the previously infected and uninfected groups. The authors must remove this conclusion - or add the appropriate statistical tests.

Response to concluding sentence in the abstract:

We realized that phrasing of the concluding sentence in the abstract may have been misleading. Please note that the final sentence was meant to be a general summary sentence on the immunogenicity data after bivalent vaccination in general, i.e. vaccine-induced immune-response parameters of the whole manuscript (shown in figures 1-5), and not specifically restricted to the part dealing with subsequent infections (figure 6 and supplementary items). To avoid misinterpretation by the general readership, we specified this more clearly, and chose a separate sentence on immunogenicity data starting with “In summary, we show that immunogenicity after BA.4/5-bivalent

vaccination differs between individuals with and without prior infection.” This is followed by a more “outlook-like” statement that our results may help to improve prediction of breakthrough infections is now placed in a separate sentence (**p. 3**).

Response to comment on interaction analyses:

We thank the reviewer for more closely specifying which type of analysis he/she had in mind. We have now conducted the interaction analysis that the reviewer has asked for, and we now put this analysis in an additional supplementary **table S4 (p. 44)**. Again, Dr. Leonardo Martinez from the Department of Epidemiology at Boston University was involved as expert statistician.

We would like to note several aspects of this analysis that are key to interpretation.

First, our primary research question is to understand whether there is a relationship between immunological factors and breakthrough infections among people with prior COVID-19 infection and, separately, among people without a history of prior COVID-19 infection. This is the rationale of how the whole manuscript is built from the start. Although we have done several other analyses that the reviewer has requested among the entire cohort, our primary question was this stratified analysis. When conducting an interaction analysis such as the one in table S4, the relationship between immunological factors and breakthrough infections in each stratification of prior COVID-19 infection status is not directly shown. Instead, the point of an interaction analysis is to assess whether a third variable (i.e., prior COVID-19 infection in this case) influences the relationship between an independent and dependent variable (i.e., immunological factors and breakthrough infection). Therefore, although we agree with the reviewer that an interaction analysis is important to show and we have performed this requested step, we do not believe it impacts or is relevant to the primary research question we set out to ask in this paper.

Second, we believe we are underpowered for an interaction analysis of this kind (see Sommet *et al.* 2023, Baranger *et al.* 2023 below). In any interaction analysis, the sample size and statistical power needed to show an interaction effect must be many times greater than that needed to assess the main effects. At times, this is 10 to 15 times greater sample size. We do not believe we have such a sample size. Therefore, we cannot be certain whether the lack of a statistical significance in most of the interaction analyses seen in **table S4** is due to no effect or due to limited sample size.

Despite these notes, we have performed this interaction analysis and added several lines of text to the methods (**p. 20**) and results section (**p. 11**) of the manuscript detailing this additional analysis.

Limitations from sample size are included in the discussion on **p. 17**. The following two references were also included (**reference 20 and 21**).

References:

Sommet N, Weissman DL, Cheutin N, Elliot AJ. How many participants do I need to test an interaction? Conducting an appropriate power analysis and achieving sufficient power to detect an interaction. *Advances in Methods and Practices in Psychological Science*. 2023 Sep;6(3):25152459231178728.

Baranger DA, Finsaas MC, Goldstein BL, Vize CE, Lynam DR, Olino TM. Power analyses for interaction effects in cross-sectional regressions. *Advances in Methods and Practices in Psychological Science*. 2023 Sep;6(3):25152459231187531.

We are interested in solving the matter satisfactorily, and we hope that we have reached a balanced and objective way of describing and discussing our data, whilst pointing out limitations and areas for future studies.

REVIEWERS' COMMENTS

Reviewer #3 (Remarks to the Author):

All of this reviewer's concerns have now been addressed. I thank the authors for their careful responses and addressing this reviewers concerns.

I have a small recommendation. I suggest the authors expand new table S4 to show all the coefficients from the model (and not just the interaction terms).

Response to Reviewer 3,

All of this reviewer's concerns have now been addressed. I thank the authors for their careful responses and addressing this reviewer's concerns.

I have a small recommendation. I suggest the authors expand new table S4 to show all the coefficients from the model (and not just the interaction terms).

We have followed the recommendations of the reviewer and have added individual parameters to table S4.